# A Python library for solving ice sheet modeling problems using Physics Informed Neural Networks, PINNICLE v1.0

Gong Cheng[1], Mansa Krishna[1], and Mathieu Morlighem[1]

[1]Department of Earth Sciences, Dartmouth College, Hanover, NH 03755, USA

**Correspondence:** Gong Cheng (gong.cheng@dartmouth.edu)

**Abstract.** Predicting the future contributions of the ice sheets to sea level rise remains a significant challenge due to our limited understanding of key physical processes (e.g., basal friction, ice rheology) and the lack of observations of critical model inputs (e.g., bed topography). Traditional numerical models typically rely on data assimilation methods to estimate these variables by solving inverse problems based on conservation laws of mass, momentum, and energy. However, these methods are not versatile and require extensive code development to incorporate new physics. Moreover, their dependence on data alignment within computational grids hampers their adaptability, especially in the context of sparse data availability in space and time. To address these limitations, we developed PINNICLE (Physics-Informed Neural Networks for Ice and CLimatE), an open-source Python library dedicated to ice sheet modeling. PINNICLE seamlessly integrates observational data and physical laws, facilitating the solution of both forward and inverse problems within a single framework. PINNICLE currently supports a variety of conservation laws, including the Shelfy-Stream Approximation (SSA), Mono-Layer Higher-Order (MOLHO) models, and mass conservation equations, for both time-independent and time-dependent simulations. The library is user-friendly, requiring only the setting of a few hyperparameters for standard modeling tasks, while advanced users can define custom models within the framework. Additionally, PINNICLE is based on the DeepXDE library, which supports widely-used machine learning packages such as TensorFlow, PyTorch, and JAX, enabling users to select the backend that best fits their hardware. We describe here the implementation of PINNICLE and showcase this library with examples across the Greenland and Antarctic ice sheets for a range of forward and inverse problems.

## 1 Introduction

Ice sheet modeling is essential for projecting future sea level rise and understanding the complex dynamics that drive ice sheet behavior under changing climate conditions (e.g., Larour et al., 2012; Aschwanden et al., 2019; Goelzer et al., 2020; Seroussi et al., 2020). These numerical models provide insights into fundamental glaciological processes, advancing our understanding of ice flow and its response to and interactions with the climate system (e.g., Schoof, 2007; Joughin et al., 2021; Morlighem et al., 2024). However, developing models that accurately capture current ice sheet behavior and mass change remains challenging, further complicated by limited observational data and an incomplete understanding of critical physical processes. Despite decades of development, significant uncertainties persist in ice sheet models, impacting the accuracy of sea-level projections (Robel et al., 2019; Edwards et al., 2019; Aschwanden et al., 2021). A major source of uncertainty in models comes from pa-

rameters that are challenging to measure directly in the field, such as the bed topography (Morlighem et al., 2017, 2020), basal conditions (Gagliardini et al., 2007; Morlighem et al., 2013; Wernecke et al., 2023), and material properties of ice, including its rheology and thermal structure (e.g., Furst et al., 2015; Colgan et al., 2021).

Traditionally, these parameters are estimated by solving inverse problems, often framed as partial differential equation (PDE) constrained optimization problems (MacAyeal, 1993; Morlighem et al., 2010; Goldberg and Sergienko, 2011). These inverse problems are difficult to solve because they require customized numerical algorithms that depend on the specific PDEs governing the system (Goldberg and Heimbach, 2013; Cheng and Lotstedt, 2020) and often demand regularization to avoid overfitting and facilitate convergence due to their inherent ill-posedness (Tikhonov, 1943). As data availability and quality continue to improve, integrating these data into traditional inverse frameworks presents new challenges, both in terms of computational demands and the complexity of the required numerical methods. This increase in data complexity calls for new approaches that can flexibly integrate observational data while remaining computationally efficient.

Recent advances in machine learning offer promising alternatives for handling the shortcomings of traditional ice sheet models, particularly when dealing with sparse or noisy data (Karniadakis et al., 2021). In particular, Physics-Informed Neural Networks (PINNs) have gained attention as a powerful tool for combining physical laws with observational data (Raissi et al., 2019; Karniadakis et al., 2021; Lu et al., 2021). Unlike conventional approaches that require extensive customization for each specific problem, PINNs allow for the flexible integration of various physical constraints, making them highly adaptable to different modeling scenarios. In glaciology, PINNs have been applied to a range of challenging modeling tasks recently, including inferring basal conditions, simulating complex ice flow dynamics, and testing novel hypotheses about physical processes in ice shelves (Riel et al., 2021; Wang et al., 2022; Jouvet and Cordonnier, 2023; Iwasaki and Lai, 2023; Cheng et al., 2024). The incorporation of physical knowledge within the neural network structure has the potential to introduce a regularizing effect, which is particularly beneficial for handling sparse and noisy observational data, supporting mesh-free modeling, and enabling flexible constraints beyond standard boundary conditions (Seo, 2024). This framework provides a balance between model complexity and accuracy, making it a robust tool for advancing ice sheet modeling (Raissi et al., 2020; Cheng et al., 2024).

In this context, we describe here PINNICLE (Physics-Informed Neural Networks for Ice and CLimatE), an open-source Python library for ice sheet modeling designed to facilitate the solution of both forward and inverse problems using PINNs. PINNICLE provides a unified framework to integrate observational data directly with the governing physical laws. Leveraging the DeepXDE library (Lu et al., 2021), PINNICLE supports machine learning platforms like TensorFlow, PyTorch, and JAX, giving users the flexibility to choose the backend that best suits their hardware and computational resources. This paper outlines the methodological framework of PINNICLE and illustrates its capabilities through applications on glaciers in the Greenland and Antarctic Ice Sheets.

## 2 Physics Informed Neural Networks

In recent years, physics-informed machine learning techniques have become increasingly popular in the field of ice sheet modeling (e.g., Riel et al., 2021; Brinkerhoff, 2022; Jouvet and Cordonnier, 2023; Bolibar et al., 2023; He et al., 2023). Ice

flow is governed by a set of PDEs derived from fundamental conservation laws. The extent to which these governing equations are satisfied in different machine learning frameworks varies across studies. For example, the approaches described in He et al. (2023); Bolibar et al. (2023); Koo et al. (2024) employ neural networks as emulators or surrogate models, learning internal relationships derived from numerical solutions of the governing PDEs, with minimal enforcement of physical constraints. In Jouvet and Cordonnier (2023), the neural network also acts as an emulator but incorporates the PDE residual directly into the training loss function, embedding physical principles into the optimization process for a more physically consistent model. Similarly, Riel et al. (2021) combines observational data with physics-inspired constraints, such as smoothness and sign of the basal drag, though without explicitly enforcing physical laws on the model's outputs.

This study takes an alternative approach by leveraging neural networks to directly learn the physical constraints and relationships governing ice dynamics, aiming for a more comprehensive integration of data and physics. The architecture of PINNICLE follows the same framework as in Raissi et al. (2019); Wang et al. (2022); Iwasaki and Lai (2023); Cheng et al. (2024), where a neural network is trained under the constraints of data misfit and PDE loss. The inputs of the neural network are the independent variables of the PDEs, which can be the spatial coordinates $(x, y, z)$ and/or the time variable $(t)$, depending on the governing equations. The outputs of the neural network are all the dependent variables of the PDEs. This architecture enables the neural network, through backpropagation, to compute the required spatial and temporal derivatives of these dependent variables for evaluating the PDEs. The loss function of the PINN consists of two parts, data and physical loss, which are both constructed using these output variables. The data loss measures the misfit between the data and the corresponding output variable at the location and time of the data acquired. The physical loss is computed by evaluating the residual of the PDEs on a set of randomly generated collocation points. Then, the training procedure is to minimize the loss function, such that the output of the neural network satisfies the governing PDEs and also matches the data provided. However, it is important to note that this formulation represents an idealized scenario. In practice, the residuals typically do not reach zero, nor does the data misfit, as discussed in Cheng et al. (2024).

## 3    Physics

PINNICLE is designed to be flexible so that diverse physical laws along with relevant observational data can be easily integrated. In ice sheet modeling, conservation of mass and momentum form the governing equation of ice dynamics. PINNICLE currently supports these laws by implementing widely used stress balance approximations. Users can readily apply or extend the framework to include additional physical constraints as required by specific modeling objectives.

### 3.1    Conservation of mass

In ice sheet modeling, the large aspect ratio of the horizontal dimensions $(x, y)$ to the vertical dimension $(z)$ leads most dynamics to occur in the $x$-$y$ plane. Therefore, it is common practice to model the ice sheet using a depth-averaged approach that effectively captures horizontal behavior, simplifying the model by considering a two-dimensional domain $(x, y)$ and focusing on the horizontal ice velocity components, denoted by $\boldsymbol{u} = (u, v)^T$.

Mass conservation is fundamental to simulating ice sheet dynamics, as it controls the variations in ice thickness over time. Since ice behaves as an incompressible fluid, the rate of change in ice thickness, $H$, is equal to the sum of the flux divergence in the horizontal direction and vertical mass exchange processes:

$$\frac{\partial H}{\partial t} + \frac{\partial (uH)}{\partial x} + \frac{\partial (vH)}{\partial y} = a, \tag{1}$$

where $a$ represents the net mass balance field from surface and basal processes, such as surface accumulation (e.g., snowfall) and losses like surface or basal melting.

## 3.2 Conservation of momentum

We describe here two approximations of the conservation of momentum that are widely used in glaciology. First, the Shelfy-Stream Approximation (SSA), provides a simplified form of the full Stokes equations by neglecting vertical shear stresses (MacAyeal, 1989). SSA is an excellent approximation of ice sheet flow in regions of fast sliding, such as ice streams, and floating ice shelves. This reduction allows us to describe the horizontal motion of ice through a system of PDEs, which balances gravitational driving forces with internal stress gradients and basal drag as follows:

$$\frac{\partial}{\partial x}\left(4\mu H \frac{\partial u}{\partial x} + 2\mu H \frac{\partial v}{\partial y}\right) + \frac{\partial}{\partial y}\left(\mu H \frac{\partial u}{\partial y} + \mu H \frac{\partial v}{\partial x}\right) - \tau_b \frac{u}{|\boldsymbol{u}|} = \rho_i g H \frac{\partial s}{\partial x}$$

$$\frac{\partial}{\partial x}\left(\mu H \frac{\partial u}{\partial y} + \mu H \frac{\partial v}{\partial x}\right) + \frac{\partial}{\partial y}\left(2\mu H \frac{\partial u}{\partial x} + 4\mu H \frac{\partial v}{\partial y}\right) - \tau_b \frac{v}{|\boldsymbol{u}|} = \rho_i g H \frac{\partial s}{\partial y} \tag{2}$$

where $s$ is the ice surface elevation, $\rho_i$ denotes ice density, $g$ is gravitational acceleration, $\tau_b$ is the basal shear stress, and $\mu$ denotes the ice viscosity, which follows Glen's flow law (Glen, 1958), reflecting the non-linear behavior of ice deformation:

$$\mu = \frac{B}{2}\left(\left(\frac{\partial u}{\partial x}\right)^2 + \left(\frac{\partial v}{\partial y}\right)^2 + \frac{1}{4}\left(\frac{\partial u}{\partial y} + \frac{\partial v}{\partial x}\right)^2 + \frac{\partial u}{\partial x}\frac{\partial v}{\partial y}\right)^{\frac{1-n}{2n}}, \tag{3}$$

where $n = 3$ is the flow-law exponent, and $B$ is a temperature dependent pre-factor (Cuffey and Paterson, 2010).

To characterize the relationship between basal shear stress and sliding velocity, we use Weertman's friction law (Weertman, 1957; Fowler, 1981; Cuffey and Paterson, 2010) as an example:

$$\tau_b = C^2 |\boldsymbol{u}|^m, \tag{4}$$

where $C$ is a spatially varying friction coefficient, and $m = 1/3$.

In regions where vertical deformation significantly influences ice flow, such as in the ice sheet interior, we incorporate a MOno-Layer Higher-Order (MOLHO) model into PINNICLE, following the approach in dos Santos et al. (2022). This formulation represents ice velocity $\boldsymbol{u}$ as the sum of a basal component $\boldsymbol{u}^b$ and a shear velocity $\boldsymbol{u}^{sh}$, modulated by a normalized depth factor $\zeta$ as follows:

$$\boldsymbol{u} = \boldsymbol{u}^b + \boldsymbol{u}^{sh}(1 - \zeta^{n+1}), \tag{5}$$

where $\zeta(z) = \frac{s-z}{H}$ scales with depth with $z$ varying between the ice base $b$ and the ice surface $s$.

Both basal and shear velocities are defined on a 2D domain, with vertical variations accounted for by the polynomial in $\zeta$. The MOLHO model extends the SSA equations by accounting for vertical deformation, expressed as:

$$\frac{\partial}{\partial x}\left(\bar{\mu}_1 H \epsilon_{11}^b + \bar{\mu}_2 H \epsilon_{11}^{sh}\right) + \frac{\partial}{\partial y}\left(\bar{\mu}_1 H \epsilon_{12}^b + \bar{\mu}_2 H \epsilon_{12}^{sh}\right) - \tau_b \frac{u_b}{|\boldsymbol{u}_b|} = \rho_i g H \frac{\partial s}{\partial x}$$

$$\frac{\partial}{\partial x}\left(\bar{\mu}_1 H \epsilon_{12}^b + \bar{\mu}_2 H \epsilon_{12}^{sh}\right) + \frac{\partial}{\partial y}\left(\bar{\mu}_1 H \epsilon_{22}^b + \bar{\mu}_2 H \epsilon_{22}^{sh}\right) - \tau_b \frac{v_b}{|\boldsymbol{u}_b|} = \rho_i g H \frac{\partial s}{\partial y}$$

$$\frac{\partial}{\partial x}\left(\bar{\mu}_2 H \epsilon_{11}^b + \bar{\mu}_3 H \epsilon_{11}^{sh}\right) + \frac{\partial}{\partial y}\left(\bar{\mu}_2 H \epsilon_{12}^b + \bar{\mu}_3 H \epsilon_{12}^{sh}\right) + \bar{\mu}_4 H u^{sh} = \frac{(n+1)}{(n+2)}\rho_i g H \frac{\partial s}{\partial x} \tag{6}$$

$$\frac{\partial}{\partial x}\left(\bar{\mu}_2 H \epsilon_{12}^b + \bar{\mu}_3 H \epsilon_{12}^{sh}\right) + \frac{\partial}{\partial y}\left(\bar{\mu}_2 H \epsilon_{22}^b + \bar{\mu}_3 H \epsilon_{22}^{sh}\right) + \bar{\mu}_4 H v^{sh} = \frac{(n+1)}{(n+2)}\rho_i g H \frac{\partial s}{\partial y}$$

where $\epsilon_{ij}^b$ and $\epsilon_{ij}^{sh}$ represent the strain rate components for basal and shear velocities, respectively. These components are detailed by

$$\epsilon_{11}^b = 4\frac{\partial u^b}{\partial x} + 2\frac{\partial v^b}{\partial y}, \qquad \epsilon_{12}^b = \frac{\partial u^b}{\partial y} + \frac{\partial v^b}{\partial x}, \qquad \epsilon_{22}^b = 2\frac{\partial u^b}{\partial x} + 4\frac{\partial v^b}{\partial y},$$

$$\epsilon_{11}^{sh} = 4\frac{\partial u^{sh}}{\partial x} + 2\frac{\partial v^{sh}}{\partial y}, \quad \epsilon_{12}^{sh} = \frac{\partial u^{sh}}{\partial y} + \frac{\partial v^{sh}}{\partial x}, \quad \epsilon_{22}^{sh} = 2\frac{\partial u^{sh}}{\partial x} + 4\frac{\partial v^{sh}}{\partial y}, \tag{7}$$

and the vertically integrated viscosities are calculated as follows:

$$\bar{\mu}_1 = \frac{1}{2}\int_b^s \mu\,\mathrm{d}z, \quad \bar{\mu}_2 = \frac{1}{2}\int_b^s \mu\left(1 - \zeta^{n+1}\right)\mathrm{d}z, \quad \bar{\mu}_3 = \frac{1}{2}\int_b^s \mu\left(1 - \zeta^{n+1}\right)^2\mathrm{d}z, \quad \bar{\mu}_4 = \frac{1}{2}\int_b^s \mu\left(\frac{n+1}{H}\zeta^n\right)^2\mathrm{d}z, \tag{8}$$

where $\mu$ is the effective viscosity defined as in Eq. (3).

The basal shear stress $\tau_b$ is defined using Weertman's friction law (Weertman, 1957), as in the SSA model. Alternative friction laws exist (Budd et al., 1979; Gagliardini et al., 2007), and PINNICLE can be easily adjusted to support these friction laws.

## 4  Data

PINNICLE supports a variety of data formats, including CSV, NetCDF, and MATLAB data files. These files can contain scattered data or structured data, such as the model struct from the Ice-sheet and Sea-level System Model (ISSM). The spatial and temporal coordinates of the data can be different between different variables, offering flexibility to accommodate any complex situations. Depending on the underlying problem to solve, users can specify the amount of data needed for training. The selected data contribute to the calculation of the data misfit component within the loss function.

**Table 1.** Data misfit functions implemented in PINNICLE.

| Key | Formula | Description |
|---|---|---|
| `MAE` | $E_{\text{MAE}}(\hat{d}, d) = \frac{1}{n} \sum_{i=1}^{n} |\hat{d}_i - d_i|$ | mean absolute error |
| `MSE` | $E_{\text{MSE}}(\hat{d}, d) = \frac{1}{n} \sum_{i=1}^{n} (\hat{d}_i - d_i)^2$ | mean square error |
| `MAPE` | $E_{\text{MAPE}}(\hat{d}, d) = \frac{100}{n} \sum_{i=1}^{n} \left| \frac{\hat{d}_i - d_i}{\hat{d}_i} \right|$ | mean absolute percentage error |
| `VEL_LOG` | $E_{\text{VLOG}}(\hat{d}, d) = \frac{1}{n} \sum_{i=1}^{n} \frac{\ln(|d_i| + \varepsilon)}{\ln(|\hat{d}_i| + \varepsilon)}$ | mean relative logarithmic error |
| `MEAN_SQUARE_LOG` | $E_{\text{MSLOG}}(\hat{d}, d) = \frac{1}{n} \sum_{i=1}^{n} \left( \ln(|\hat{d}_i| + 1) - \ln(|d_i| + 1) \right)^2$ | mean squared logarithmic error |

To evaluate the data misfit, PINNICLE provides multiple metrics, integrating both the built-in data misfit functions from the DeepXDE package (Lu et al., 2021) and commonly employed functions from other machine learning libraries. Table 1 summarizes the data misfit functions currently available in PINNICLE. In the table, $d$ represents the predicted solution from PINNICLE, $\hat{d}$ denotes the 'reference solution' from the data, and $\varepsilon = 2.2204 \times 10^{-16}$ corresponds to the machine epsilon for double precisions. For advanced users, PINNICLE also provides a flexible interface to define custom misfit functions to specific modeling needs.

## 5   Neural Networks

The fundamental neural network architecture implemented in PINNICLE is the fully connected neural network (FNN), which takes spatial and temporal coordinates as input, and the dependent variables of the PDEs as output, through one single neural network. This architecture allows the neural network to implicitly capture the internal relationship among the dependent variables, and has been widely used in many applications (Raissi et al., 2019; Iwasaki and Lai, 2023; Wang et al., 2022; Lu et al., 2021; Karniadakis et al., 2021; Teisberg et al., 2021). However, in some cases, the training process may not converge, as the relationship between some dependent variables may be too complex for the neural network to capture (Cheng et al., 2024). To address these problems when they occur, we also provide a parallel fully connected neural network (PFNN) (Lu et al., 2021). The PFNN uses multiple FNNs, each for one dependent variable with the same independent variables as input. In this case, each neural network only needs to learn its corresponding dependent variable from the data and the PDE constraints.

By default, we use the hyperbolic tangent function as the activation function for the neural network. To accommodate different physical problems, PINNICLE applies min-max normalization before the input layer and a min-max denormalization after the output layer. These normalization processes are strictly limited to the inputs and outputs of the neural network, while

the governing PDE residuals and data misfit terms are all defined and calculated in the International System of Units (SI). Consequently, there is no need for the user to scale the physical equations or data misfit functions themselves.

While glacier ice typically exhibits low-pass filtering behavior, some variables, such as surface elevation, contain high-frequency components. To capture these variations accurately, we implemented Fourier Feature Transform (FFT) (Tancik et al., 2020; Wang et al., 2021) in PINNICLE. Assuming the dimension of the input variables $x$ is $d$, the FFT for $m$ features is written as $f(x) = [\cos(\mathbf{B}x), \sin(\mathbf{B}x)]^T$, where $\mathbf{B} \in \mathbb{R}^{m \times d}$ is sampled from a Gaussian distribution $\mathcal{N}(0, \sigma)$, the functions $\cos(\cdot)$ and $\sin(\cdot)$ are applied coordinate-wise on the vector $\mathbf{B}x$. This transformation is applied between the min-max normalization and the input layer of the neural network.

## 6 Loss function

The loss function in PINNICLE integrates both data misfit and physical constraints. The general form of the loss function is given by:

$$\mathcal{L} = \mathcal{L}_d + \mathcal{L}_\varphi, \tag{9}$$

where $\mathcal{L}_d$ and $\mathcal{L}_\varphi$ represent the weighted sum of data misfit and PDE residual, respectively. The specific formulation of the loss function depends on the underlying physical problem, the data, and the misfit functions (Table 1) employed. Examples of detailed loss function formulations are provided in Section 8, specifically in Equations (10), (11), and (12).

Physical variables in glaciological applications span multiple orders of magnitude, for example, ice velocity is typically around $10^{-5}$ m/s, ice thickness around $10^3$ m, and driving stress of a glacier is approximately $10^5$ Pa. These differences can lead to imbalanced contributions in the optimization process, where certain variables dominate the loss function while others are underrepresented. To address this issue, PINNICLE applies weights to each individual term in the loss function, ensuring balanced contributions from different data sources and governing equations. The impact of weighting strategies on ice sheet modeling problems has been examined in previous studies (Iwasaki and Lai, 2023; Cheng et al., 2024). While no theoretical guidelines exist for determining optimal weights, an extensive set of over 15,000 numerical experiments described in Cheng et al. (2024), including an L-curve analysis, demonstrated that the best performance is achieved when the contributions of all terms are weighted to approximately the same order of magnitude (i.e., around order 1). Based on this empirical finding, PINNICLE assigns default weights for each term in the loss function accordingly. The default weights and their corresponding typical values are provided in Table 2. For greater flexibility, users can modify the weights to meet their specific modeling requirements. Examples of loss function construction and weight selection strategies are provided in Section 8.

## 7 Structure of the package

PINNICLE is configured by the user based on a nested dictionary to define the neural network architecture, governing equations, computational domains, data, and other experiment-specific settings. This design ensures flexibility for incorporating new functionalities, allowing the model to evolve alongside future advances in physics-informed machine learning. To ensure

**Table 2.** Default weights and typical values of physical variables in PINNICLE

| name | weight | default value | variable | typical value of the variable |
|---|---|---|---|---|
| Velocity | $\gamma_{\boldsymbol{u}}$ | $10^{-8} \times (31536000^2) \text{ m}^{-2}\,\text{s}^2$ | $u, v, \|\boldsymbol{u}\|$ | $10^4 \text{ m yr}^{-1}$ |
| Thickness | $\gamma_H$ | $10^{-6} \text{ m}^{-2}$ | $H$ | $10^3 \text{ m}$ |
| Surface elevation | $\gamma_s$ | $10^{-6} \text{ m}^{-2}$ | $s$ | $10^3 \text{ m}$ |
| Mass balance | $\gamma_a$ | $31536000^2 \text{ m}^{-2}\,\text{s}^2$ | $a$ | $1 \text{ m yr}^{-1}$ |
| Friction coefficient | $\gamma_C$ | $10^{-8} \text{ Pa}^{-1}\,\text{m}^{1/3}\,\text{s}^{-1/3}$ | $C$ | $10^4 \text{ Pa}^{1/2}\text{m}^{-1/6}\text{s}^{1/6}$ |
| Rheology pre-factor | $\gamma_B$ | $10^{-18} \text{ Pa}^{-2}\,\text{s}^{-2/3}$ | $B$ | $10^9 \text{ Pa s}^{1/3}$ |
| Driving stress | $\gamma_{\boldsymbol{\tau}}$ | $10^{-10} \text{ Pa}^{-2}$ | $\rho g H \|\nabla s\|$ | $10^5 \text{ Pa}$ |
| Dynamic thinning | $\gamma_{H/t}$ | $10^{10} \text{ m}^{-2}\,\text{s}^2$ | $\frac{\partial H}{\partial t}$ | $10^3 \text{ m}/31536000 \text{ s}$ |

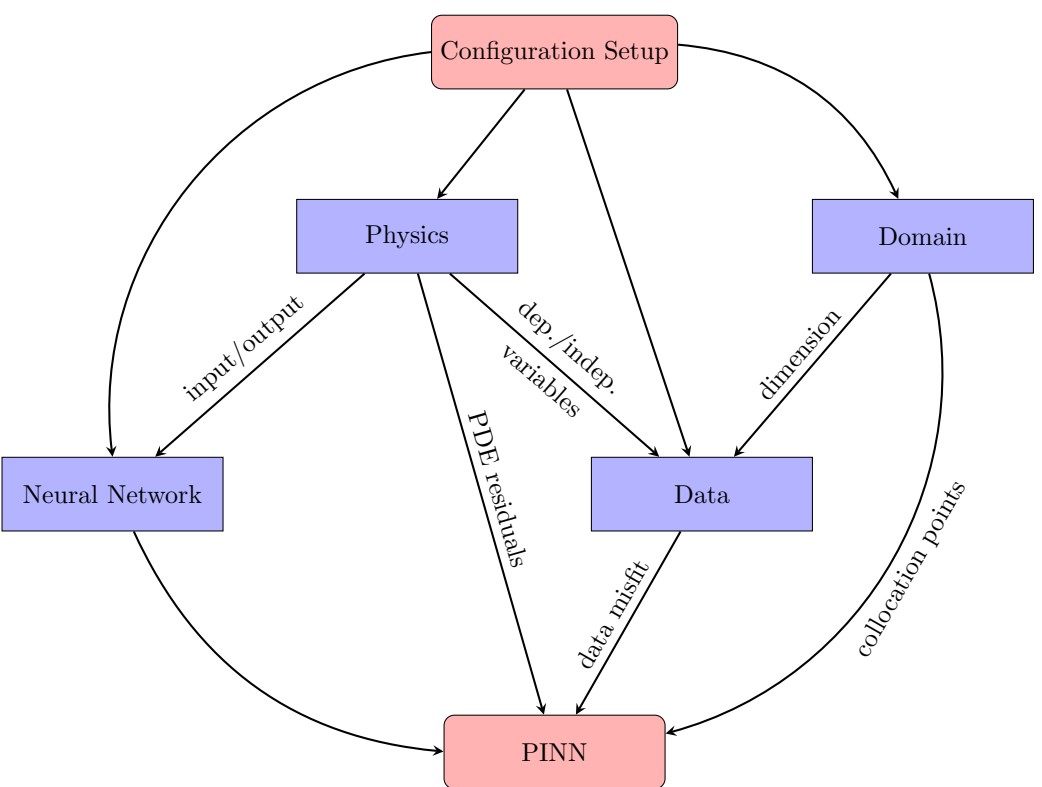

**Figure 1.** Flowchart illustrating the structure of PINNICLE, with arrows representing the flow of information between modules.

reproducibility, all user-defined configurations are saved in a JSON file, which can be reloaded to replicate simulations with the same parameters.

The core structure of PINNICLE is built upon five key modules: Physics, Neural Network, Data, Domain, and PINN. The interconnected structure of these modules is depicted in Figure 1. The Physics module constructs the PDE constraints, gathering independent and dependent variables from the governing equations. These variables are unified to establish a well-defined mapping between inputs and outputs for the PINN framework. The module assigns these variables to the Neural Network and Data modules, enabling integration of physics-based constraints into the computational pipeline.

The Neural Network module constructs the architecture for the neural network based on user configurations. It supports several popular machine learning libraries, including TensorFlow (Abadi et al., 2015), PyTorch (Paszke et al., 2019), and JAX (Bradbury et al., 2018). To enable seamless transitions between these libraries without altering the codebase, the module employs the DeepXDE framework (Lu et al., 2021) as its backend. PINNICLE supports Python 3.8 or higher and requires minimal library versions, including TensorFlow 2.11.0, PyTorch 2.5, and JAX 0.4, ensuring compatibility with current machine learning standards.

The Data module manages data integration by loading datasets and assigning them to the model with specific loss functions as described in Section 4. Users can control the volume of training data, enabling the PINNICLE to handle forward or inverse problems based on the combination of data and PDEs provided. To prevent overfitting and generalization, PINNICLE employs a random sampling strategy to automatically load data from the data file depending on their types, which also acts as a form of implicit regularization in the optimization problem. For mesh data (e.g., ISSM or NetCDF), it applies uniform random sampling, while scattered data is downsampled using Cartesian grids to maintain spatial coverage and avoid over-clustering before random sampling.

The Domain module defines the computational domain by generating a polygon based on a list of user-defined vertices. Using Hammersley sequence sampling (Wong et al., 1997), it produces quasi-random collocation points within the domain. These points are then utilized during training to evaluate residuals of the governing PDEs.

After configuring all four modules, the PINN module integrates them using the DeepXDE package, which provides a comprehensive framework for compiling and training PINNs. By leveraging DeepXDE's built-in capabilities, PINNICLE streamlines the training process, allowing users to focus on model setup and interpretation rather than low-level implementation details.

## 8 Example of Applications

### 8.1 Example 1: an SSA inverse problem on Helheim Glacier

In this example, we use PINNICLE to reproduce results of the inverse problem described in Cheng et al. (2024). The problem involves solving the 2D SSA equation outlined in Eq. (2) for the fast-flowing region of Helheim Glacier in Southeast Greenland. We use a 6-layer fully connected neural network, with 20 neurons per layer. A schematic representation of the problem, including the data used for training, is shown in Figure 2 and the corresponding Python implementation is provided in Listing 1. We configure PINNICLE to randomly sample $N_{\mathbf{u}} = N_H = N_s = 4000$ data points for each input variables (line 19 to 22), including ice velocities, surface elevation, ice thickness, along with $N_{\varphi} = 9000$ collocation points over the entire basin of

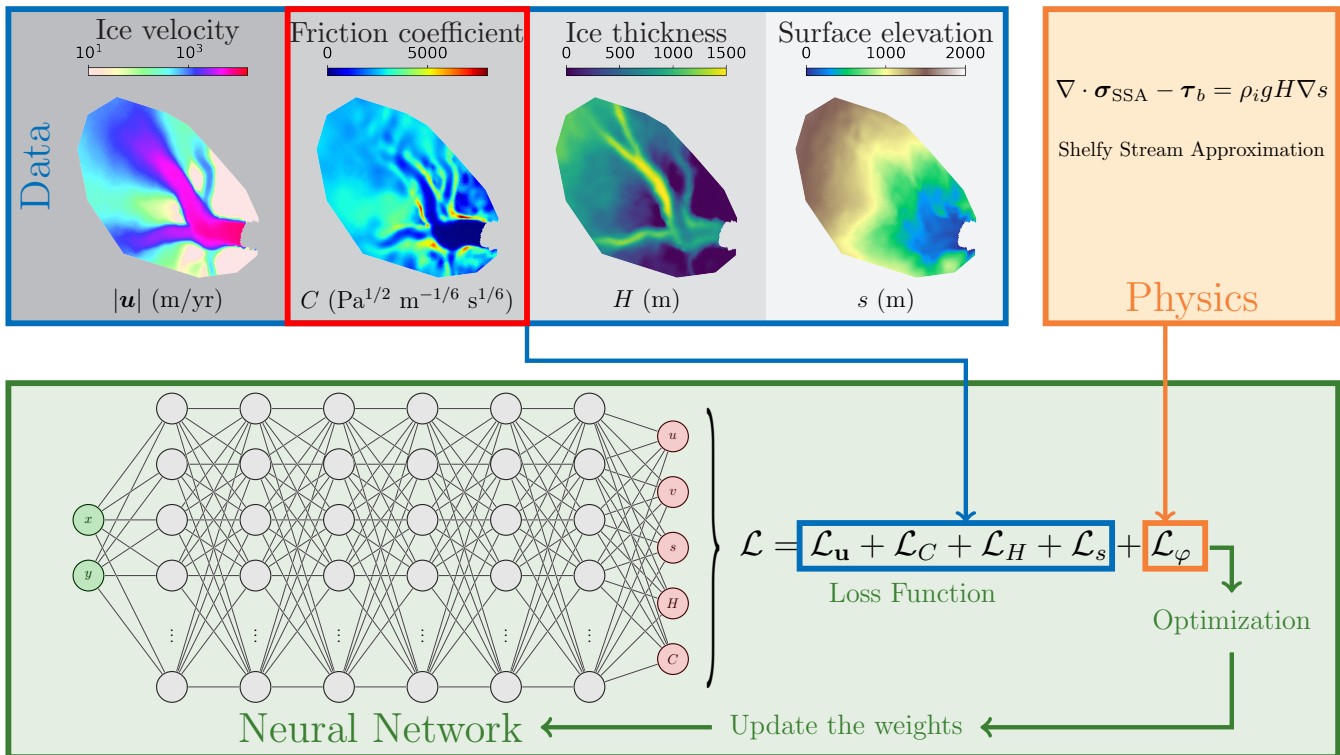

**Figure 2.** The PINNICLE framework for Example 1: inverse problem on Helheim Glacier. The training data include ice velocity, ice thickness, and surface elevation. The friction coefficient $C$, highlighted in the red box, represents the numerical solution from ISSM and is used only on the domain boundary as a boundary condition. It also serves as the 'reference solution' for comparison. The governing equation in the box of Physics is the vector form of the SSA in Eq. (2).

Helheim Glacier for evaluating the PDE residuals (line 12 to 13). The unknown friction coefficient is inferred by specifying `"C":None` in the data section (line 20), ensuring that only boundary data is used to constrain the inversion. It is important to note that the friction coefficient within the red box in Figure 2 is not included in the training processes but is retained as the 'reference solution' for validation purposes in Figures 3(b) and 3(f).

Once the setup is complete, PINNICLE assembles the loss function based on these user-defined inputs. In this case, the complete formulation of the loss function is written as

$$\mathcal{L} = \underbrace{\frac{\gamma_{\mathbf{u}}}{N_{\mathbf{u}}} \sum_{i=1}^{N_{\mathbf{u}}} \left( (\hat{u}_i - u_i)^2 + (\hat{v}_i - v_i)^2 \right)}_{\mathcal{L}_{\mathbf{u}}} + \underbrace{\frac{\gamma_H}{N_H} \sum_{i=1}^{N_H} \left( \hat{H}_i - H_i \right)^2}_{\mathcal{L}_H} + \underbrace{\frac{\gamma_s}{N_s} \sum_{i=1}^{N_s} (\hat{s}_i - s_i)^2}_{\mathcal{L}_s} + \underbrace{\frac{\gamma_C}{N_C} \sum_{i=1}^{N_C} \left( \hat{C}_i - C_i \right)^2}_{\mathcal{L}_C}$$

$$+ \underbrace{\frac{\gamma_{\boldsymbol{\tau}}}{N_{\varphi}} \sum_{i=1}^{N_{\varphi}} |\nabla \cdot \boldsymbol{\sigma}_{\text{SSA}} - \boldsymbol{\tau}_b - \rho_i g H \nabla s|^2}_{\mathcal{L}_{\varphi}}, \tag{10}$$

```python
1: import pinnicle
2:
3: # General parameters
4: hp = {}
5: hp["epochs"] = 100000
6:
7: # NN
8: hp["num_neurons"] = 20
9: hp["num_layers"] = 6
10:
11: # domain
12: hp["shapefile"] = "Helheim.exp"
13: hp["num_collocation_points"] = 9000
14:
15: # physics
16: hp["equations"] = {"SSA":{}}
17:
18: # data
19: issm = {}
20: issm["data_size"] = {"u":4000, "v":4000, "s":4000, "H":4000, "C":None}
21: issm["data_path"] = "Helheim.mat"
22: hp["data"] = {"ISSM":issm}
23:
24: # create experiment
25: experiment = pinnicle.PINN(hp)
26: experiment.compile()
27:
28: # Train
29: experiment.train()
```

**Listing 1.** Python code of Example 1: inverse problem on Helheim Glacier.

where the weights $\gamma_{(\cdot)}$ are the default weights in PINNICLE as shown in Table 2, and $N_C = 541$ is the number of boundary points used for constraining friction coefficient $C$. The terms $\mathcal{L}_{\mathbf{u}}$, $\mathcal{L}_H$, and $\mathcal{L}_s$ are the weighted MSE of the data misfit for ice velocity $|\mathbf{u}|$, ice thickness $H$, and surface elevation $s$, respectively. The term $\mathcal{L}_C$ accounts for the MSE of the boundary condition imposed on the friction coefficient. However, it is possible treat $C$ as a free parameter in the inversion. In such cases, users can exclude $C$ from the dataset by removing the entry `"C":None` from `issm["data_size"]` configuration (e.g., line 20 in Listing 1). PINNICLE will then automatically omit the $\mathcal{L}_C$ term from the loss function in Eq. (10). The PDE residual, denoted as $\mathcal{L}_\varphi$, is expressed in the vector form of the SSA in Eq. (2) and is weighted according to the driving stress to ensure balanced contributions in the loss function.

After training for 100,000 epochs (line 5 in Listing 1), the solution is presented in Figure 3. The root mean square error (RMSE) for each variable comparing to the 'reference solution' is as follows: velocity at 273.86 m/a, surface elevation at 22.21 m, ice thickness at 29.07 m, and friction coefficient at 1101.96 $\mathrm{Pa}^{1/2}\mathrm{m}^{-1/6}\mathrm{s}^{1/6}$. These results closely align with those reported in Cheng et al. (2024), with similar spatial patterns of errors. However, our errors are approximately twice as large, due to training for only one-tenth of the epochs compared to Cheng et al. (2024). Despite this, the overall consistency demonstrates the capability of PINNICLE to reproduce and approximate inverse problem results effectively.

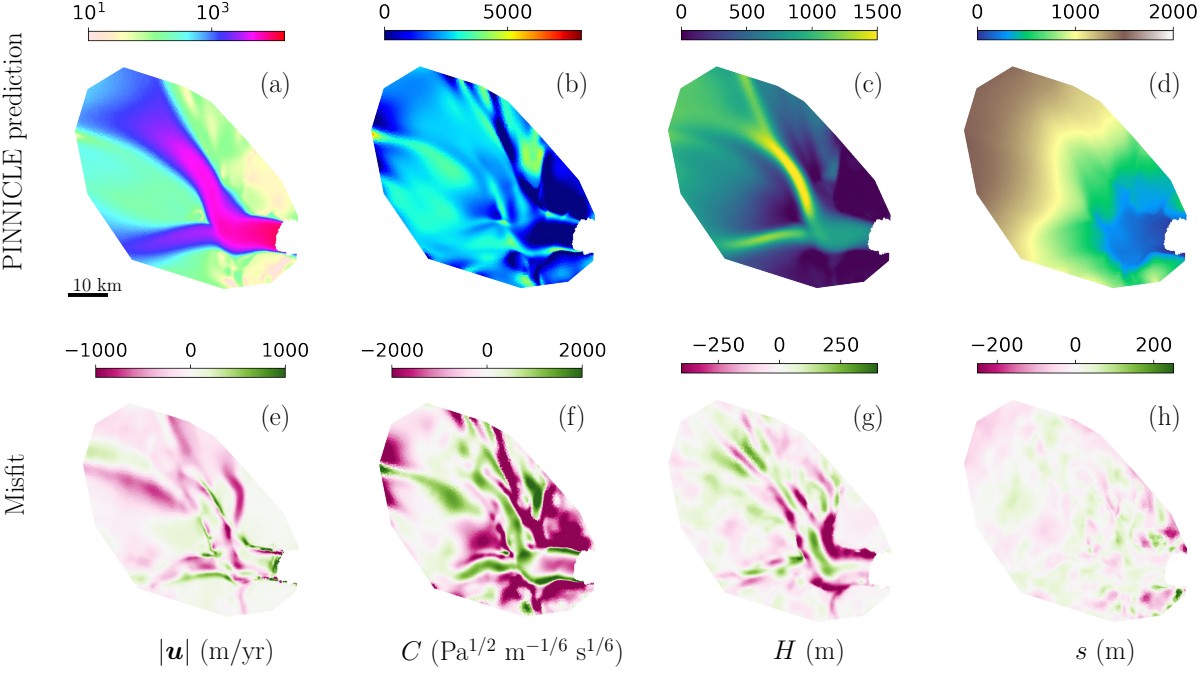

**Figure 3.** The PINNICLE predictions and the misfit of Example 1. (a)-(d) The predictions of surface velocity, friction coefficient, ice thickness, and surface elevation. (e)-(h) The misfit between the 'reference solution' in Figure 2 and the corresponding PINNICLE predictions in (a)-(d).

## 8.2 Example 2: Simultaneous Inference of Basal Friction and Ice Rheology for Pine Island Glacier

This second example demonstrates PINNICLE's capability to simultaneously infer two spatially varying parameters: the basal friction coefficient beneath grounded ice and the ice rheology, described by the flow-law pre-factor, within the floating ice shelf. We employ the SSA in Eq. (2) on Pine Island Glacier, Antarctica. To capture ice rheology variations, we incorporate a spatially varying pre-factor $B$ in Eq. (3). The Python implementation of this example is provided in Listing 2. The neural network architecture consists of a 6-layer fully connected network with 40 neurons per layer. To effectively capture high-frequency variations, we apply a Fourier Feature Transformation with $\sigma = 10$ and $m = 30$, as described in Section 5 (line 10 to 12 in Listing 2). Given the larger domain compared to Example 1 in Section 8.1, we utilize $N_{\mathbf{u}} = N_H = N_s = 8000$ data points and $N_{\varphi} = 18000$ collocation points to ensure comprehensive spatial coverage and resolution. The training dataset comprises surface velocity, surface elevation, and ice thickness data across the entire domain from `PIG.mat`. To constrain the model, we impose Dirichlet boundary conditions: setting $C = 0$ on the floating ice shelf with $N_C = 4000$ and prescribing $B = 1.41 \times 10^8 \text{Pa s}^{1/3}$ for grounded ice with $N_B = 4000$, corresponding to ice at -10°C (Cuffey and Paterson, 2010). These two boundary conditions are imposed using a separate data file `BC.mat`, which contains scatter data points of $C$ and $B$, and their corresponding coordinates.

In this case, the loss function is formulated as

$$\mathcal{L} = \mathcal{L}_{\mathbf{u}} + \mathcal{L}_H + \mathcal{L}_s + \underbrace{\frac{\gamma_C}{N_C} \sum_{i=1}^{N_C} \left( \hat{C}_i - C_i \right)^2}_{\mathcal{L}_C} + \underbrace{\frac{\gamma_B}{N_B} \sum_{i=1}^{N_B} \left( \hat{B}_i - B_i \right)^2}_{\mathcal{L}_B} + \mathcal{L}_{\varphi}, \tag{11}$$

where $\mathcal{L}_{\mathbf{u}}$, $\mathcal{L}_H$, $\mathcal{L}_s$, and $\mathcal{L}_{\varphi}$ have the same formulation as in Eq. (10). The additional terms $\mathcal{L}_C$ and $\mathcal{L}_B$ account for the data misfit of the friction coefficient $C$ on the floating ice shelf and the ice rheology pre-factor $B$ on the grounded ice, respectively.

After 1,000,000 training epochs, the results are presented in Figure 4. The RMSE for each variable, compared to the 'reference solution' is as follows: surface velocity at 173.13 m/a, surface elevation at 18.28 m, ice thickness at 24.38 m, basal friction coefficient at 1242.05 $\text{Pa}^{1/2}\text{m}^{-1/6}\text{s}^{1/6}$, and ice rheology pre-factor at $3.81 \times 10^7 \text{Pa s}^{1/3}$. The magnitudes of these errors are consistent with those observed in Example 1. Larger misfits are observed in two areas: the friction coefficient in slow-moving regions and the rheology near the ice front. Despite these localized discrepancies, the overall performance demonstrates PINNICLE's capacity to simultaneously infer multiple variables.

## 8.3 Example 3: Time-Dependent Forward Modeling of Helheim Glacier

In this final example, we demonstrate PINNICLE's capability to solve a transient problem by modeling the evolution of ice thickness over time using the mass transport equation in Eq. (1). Specifically, we simulate the dynamics of Helheim Glacier from 2008 to 2009. The framework for this example is illustrated in Figure 5, and the corresponding Python implementation is provided in Listing 3. Since the problem is defined as time-dependent (line 8 to 10), PINNICLE automatically constructs a spatiotemporal domain and sets the input variables of the neural network to include spatial coordinates $x, y$ and time $t$.

```python
1:  import pinnicle
2:
3:  # hyperparameters
4:  hp = {}
5:  hp["epochs"] = 1000000
6:
7:  # NN
8:  hp["num_neurons"] = 40
9:  hp["num_layers"]  = 6
10: hp["fft"] = True
11: hp['sigma'] = 10
12: hp['num_fourier_feature'] = 30
13:
14: # domain
15: hp["shapefile"] = "PIG.exp"
16: hp["num_collocation_points"] = 18000
17:
18: # physics
19: hp["equations"] = {"SSA_VB": {}}
20:
21: # data
22: issm = {}
23: issm["data_size"] = {"u":8000, "v":8000, "s":8000, "H":8000}
24: issm["data_path"] = "PIG.mat"
25: BC = {}
26: BC["data_size"] = {"C":4000, "B":4000}
27: BC["data_path"] = "BC.mat"
28: BC["source"] = "mat"
29: hp["data"] = {"ISSM":issm, "BC":BC}
30:
31: # create experiment
32: experiment = pinnicle.PINN(hp)
33: experiment.compile()
34:
35: # Train
36: experiment.train()
```

**Listing 2.** Python code of example 2: Inferring basal friction and ice rheology for Pine Island Glacier.

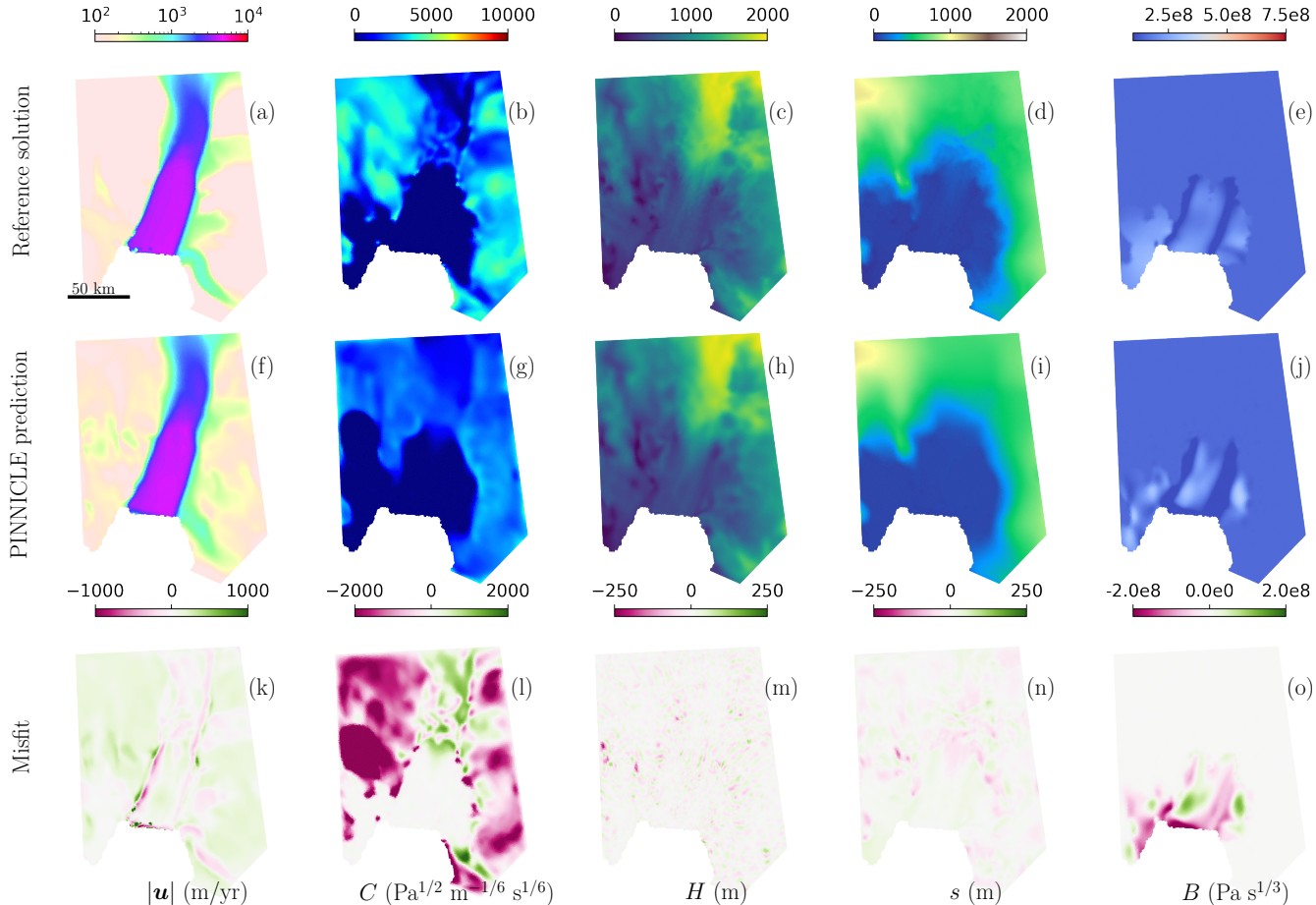

**Figure 4.** The comparison of the 'reference solution', PINNICLE predictions and misfit of example 2, to infer the basal friction coefficient and ice rheology simultaneously for Pine Island Glacier. (a)-(e) The 'reference solution' of surface velocity, friction coefficient, ice thickness, surface elevation, and the temperature dependent pre-factor. (f)-(j) The PINNICLE predictions of surface velocity, friction coefficient, ice thickness, surface elevation, and the temperature dependent pre-factor. (k)-(o) The misfit between the 'reference solution' and the corresponding PINNICLE predictions.

To drive the forward model, we provide a time series of ice velocity and surface mass balance at $0.1$ year intervals ($N_t = 11$), with each time slice containing $N_{\mathbf{u}} = N_a = 3000$ data points for each variable. The initial condition of the ice thickness is specified at $t = 2008$ also using $N_H = 3000$ data points. These data are extracted from the transient simulation results of
Cheng et al. (2022). Additionally, we randomly select $N_\varphi = 10000$ collocation points distributed across the spatiotemporal

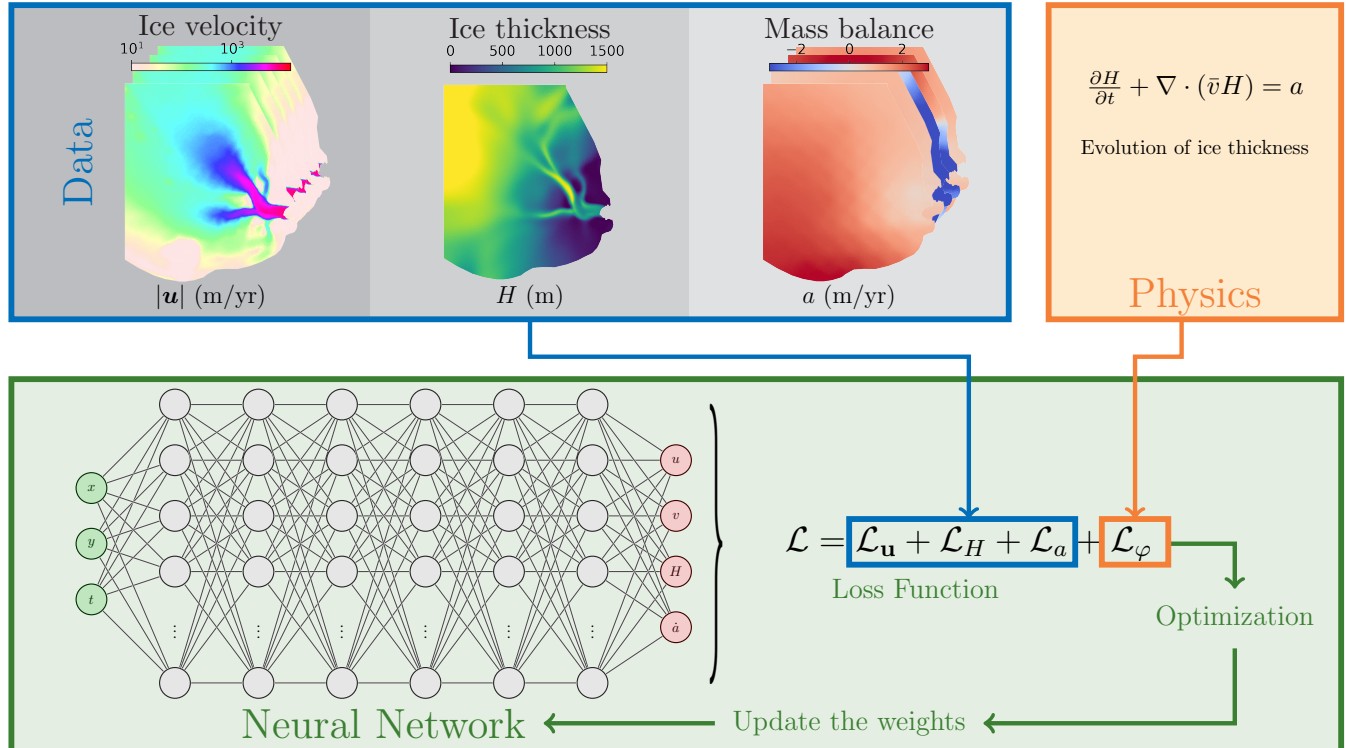

**Figure 5.** The PINNICLE framework for Example 3: the forward time-dependent problem. The training data consist of the initial ice thickness, a time series of ice velocity and apparent mass balance. Notably, the neural network inputs are automatically adapted to incorporate both spatial and temporal variables.

domain to enforce the governing equation. The loss function is written as

$$
\mathcal{L} = \frac{\gamma_{\mathbf{u}}}{N_{\mathbf{u}} N_t} \sum_{j=1}^{N_t} \sum_{i=1}^{N_{\mathbf{u}}} \left( (\hat{u}_{i,j} - u_{i,j})^2 + (\hat{v}_{i,j} - v_{i,j})^2 \right) + \frac{\gamma_H}{N_H} \sum_{i=1}^{N_H} \left( \hat{H}_i - H_i \right)^2 + \frac{\gamma_a}{N_a N_t} \sum_{j=1}^{N_t} \sum_{i=1}^{N_a} (\hat{a}_{i,j} - a_{i,j})^2
$$
$$
\underbrace{\phantom{\frac{\gamma_{\mathbf{u}}}{N_{\mathbf{u}} N_t} \sum_{j=1}^{N_t} \sum_{i=1}^{N_{\mathbf{u}}} \left( (\hat{u}_{i,j} - u_{i,j})^2 + (\hat{v}_{i,j} - v_{i,j})^2 \right)}}_{\mathcal{L}_{\mathbf{u}}} \quad \underbrace{\phantom{\frac{\gamma_H}{N_H} \sum_{i=1}^{N_H}}}_{\mathcal{L}_H} \quad \underbrace{\phantom{\frac{\gamma_a}{N_a N_t} \sum}}_{\mathcal{L}_a}
$$
$$
+ \underbrace{\frac{\gamma_{H/t}}{N_\varphi} \sum_{i=1}^{N_\varphi} \left| \frac{\partial H}{\partial t} + \nabla \cdot (\bar{v} H) - a \right|^2}_{\mathcal{L}_\varphi}, \tag{12}
$$

where $\mathcal{L}_{\mathbf{u}}$ and $\mathcal{L}_a$ represent the weighted data misfit functions of ice velocity and mass balance across the entire spatial and temporal domain, $\mathcal{L}_H$ accounts for the misfit in the initial ice thickness, and $\mathcal{L}_\varphi$ corresponds to the PDE residual of the mass

conservation in Eq. (1).

After training for 800,000 epochs using a fully connected neural network with 6 layers and 32 neurons per layer, the results at the initial time step ($t = 2008$) are shown in Figure 6, while those at the final time step ($t = 2009$) are presented in Figure 7. The RMSE for each variable over the entire simulation period (2008–2009), compared to the 'reference solution' are as

```
1:  import pinnicle
2:  import numpy as np
3:  # hyperparameters
4:  hp = {}
5:  hp["epochs"] = 800000
6:
7:  # time dependent problem
8:  hp["time_dependent"] = True
9:  hp["start_time"]     = 2008
10: hp["end_time"]       = 2009
11:
12: # NN
13: hp["num_neurons"] = 32
14: hp["num_layers"]  = 6
15:
16: # domain
17: hp["shapefile"] = "Helheim_Basin.exp"
18: hp["num_collocation_points"] = 10000
19:
20: # physics
21: hp["equations"] = {"Mass transport":{}}
22:
23: # data
24: hp["data"] = {}
25: for t in np.linspace(2008,2009,11):
26:     issm = {}
27:     if t == 2008:
28:         issm["data_size"] = {"u":3000, "v":3000, "a":3000, "H":3000}
29:     else:
30:         issm["data_size"] = {"u":3000, "v":3000, "a":3000, "H":None}
31:
32:     issm["data_path"]        = "Helheim_Transient_" + "%g"%t + ".mat"
33:     issm["default_time"]     = t
34:     issm["source"]           = "ISSM"
35:     hp["data"]["ISSM"+"%g"%t] = issm
36:
37: # create experiment
38: experiment = pinnicle.PINN(hp)
39: experiment.compile()
40:
41: # Train
42: experiment.train()
```

**Listing 3.** Python code of example 3: time-dependent problem of Helheim Glacier.

follows: surface velocity at 186.18 m/a, surface mass balance at 0.03 m/a, and ice thickness at 53.11 m. These errors remain within the same order of magnitude as the other two examples, demonstrating PINNICLE's performance in solving transient problems.

## 9  Performance

We present the performance of the examples in Table 3. All experiments were conducted on the Texas Advanced Computing Center Lonestar6 system, using nodes equipped with NVIDIA A100 GPUs with 40 GB of high bandwidth memory each. The reported wall time (in hours) represents the average of five identical runs for each numerical experiment. The computational cost is influenced by the number of epochs, neural network parameters, data points, and collocation points, among other factors. It is essential to recognize that problems similar to Examples 1 and 3 typically can be resolved more quickly with traditional numerical models, such as ISSM (Larour et al., 2012), often within just a few minutes on multi-core CPUs to achieve comparable accuracy (Cheng et al., 2024, 2022). However, solving complex mixed inverse problems, like the one presented in Example 2, using traditional adjoint methods is non-trivial, since it would include the derivation of new adjoint equations with respect to the two unknown variables, simultaneously. Furthermore, we highlight that PINNICLE's current computational performance has not been fully optimized yet, and rapid advances in hardware technology will help further increase its performance. For example, migrating our experiments from NVIDIA V100 GPUs (Cheng et al., 2024) to NVIDIA A100 GPUs has led to at least a twofold reduction in computation time.

Additionally, direct comparisons between machine learning frameworks (on GPU) and traditional data assimilation methods (on CPU) remain limited even in broader climate science community. Lai et al. (2024), for example, points out critical differences between PINNs and conventional ensemble-based data assimilation methods, such as Ensemble Kalman Filters (Bauer et al., 2015). PINNs offer the distinct advantage of simultaneously addressing forward and inverse problems, significantly reducing computational demands compared to ensemble methods, which require multiple simulation runs. Conversely, ensemble methods, despite being computationally intensive, provide explicit uncertainty quantification through ensemble spread. Relative to classical adjoint methods, PINNs demonstrate enhanced robustness to noisy observational data, effectively resolving inversion problems with smoother, physically consistent solutions (Wang et al., 2022; Cheng et al., 2024).

For the current version of PINNICLE, all spatial, temporal, and parameter derivatives are computed using reverse-mode automatic differentiation (AD) as implemented in the DeepXDE backend. Although it has been shown that using forward-mode AD for spatial derivatives (e.g., with respect to $t$, $x$, and $y$) and reverse-mode AD for network parameters can significantly improve memory and computational efficiency Cho et al. (2023), the current version of DeepXDE does not yet support this separable AD approach. We are actively following the ongoing development of this feature and plan to integrate it into PINNICLE as soon as it becomes available. We anticipate that the incorporation of this strategy in future versions of PINNICLE will further enhance the scalability and efficiency, particularly for large scale simulations.

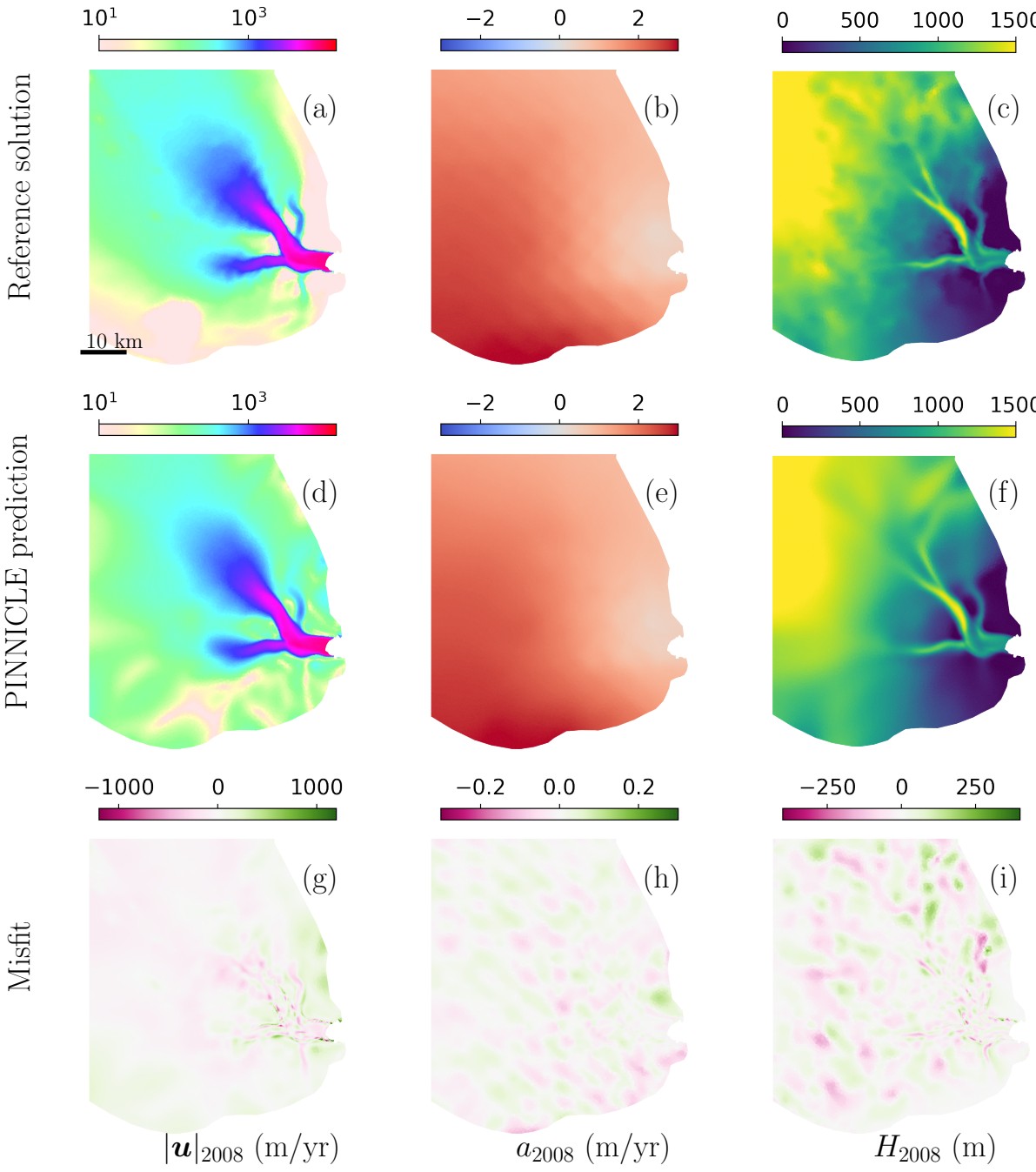

**Figure 6.** The comparison of the 'reference solution', PINNICLE predictions and misfits of the transient simulation in example 3 at the initial time step $t = 2008$. (a)-(c) The 'reference solution' of surface velocity, mass balance, and ice thickness. (d)-(f) The PINNICLE predictions. (g)-(i) The misfit between the 'reference solution' and the corresponding PINNICLE predictions.

**Table 3.** Performance of the examples in PINNICLE

| Name | Equations | Neural Network | Epochs | # Parameters | # Data | # Collocation points | Average wall time |
|---|---|---|---|---|---|---|---|
| Example 1 | SSA | 6×20 | $10^5$ | 2,286 | 4,000 | 9,000 | 0.48 h |
| Example 2 | SSA_VB | 6×40 | $10^6$ | 10,886 | 8,000 | 18,000 | 14.97 h |
| Example 3 | Mass transport | 6×32 | $8 \times 10^5$ | 5540 | 33,000 | 10,000 | 4.22 h |

## 10 Conclusions

In this study, we introduced PINNICLE, a flexible and robust framework designed to solve a wide range of glaciological problems using Physics-Informed Neural Networks. PINNICLE integrates observational data with physical laws, enabling both inverse and forward modeling across diverse spatial and temporal scales. The framework demonstrates consistent performance in inferring spatially and temporally varying parameters. PINNICLE's flexibility allows users to customize neural network architectures, incorporate various types of data, and apply different physical constraints, making it suitable for complex modeling tasks in glaciology. Additionally, we emphasize that the PINNICLE framework is not intended to replace traditional numerical methods for solving standard forward or inverse problems. Instead, it is best viewed as a complementary tool, especially useful in scenarios involving the integration of diverse physical processes and the exploration of innovative scientific concepts. PINNICLE's ability to seamlessly integrate observational data with physical laws within a unified framework enhances flexibility and can significantly simplify the implementation and evaluation of new physical models. Future developments will focus on incorporating more advanced neural network architectures, optimization techniques, and additional physical constraints to further enhance accuracy and computational efficiency. PINNICLE's modular design ensures adaptability to evolving machine learning methodologies and glaciological challenges, positioning it as a valuable tool for the cryosphere community.

*Code and data availability.* The source code and development history are hosted on GitHub at https://github.com/ISSMteam/PINNICLE. The specific version of PINNICLE used in this study, including all examples, input data used for training, and neural network weights after training, has been archived on Zenodo and is available at: https://doi.org/10.5281/zenodo.15643042 (Cheng et al., 2025). All examples mentioned in this study are organized in the folder: `PINNICLE/examples/`. The code used in this work is available as a Python package on PyPI. It can be installed using: `pip install pinnicle`. This software is licensed under the GNU Lesser General Public License v2 (LGPLv2).

*Author contributions.* GC, MK and MM designed the study. GC did the numerical computations. GC wrote the manuscript with input from MK and MM.

*Competing interests.* The authors declare that they have no conflict of interest.

*Acknowledgements.* This work was supported by the National Science Foundation #2118285 and #2147601. GC acknowledges support from the Novo Nordisk Foundation under the Challenge Programme 2023 (Grant NNF23OC00807040). The authors gratefully acknowledge the data and models obtained from projects supported by the Heising-Simons Foundation under Grants 2019-1161 and 2021-3059.

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

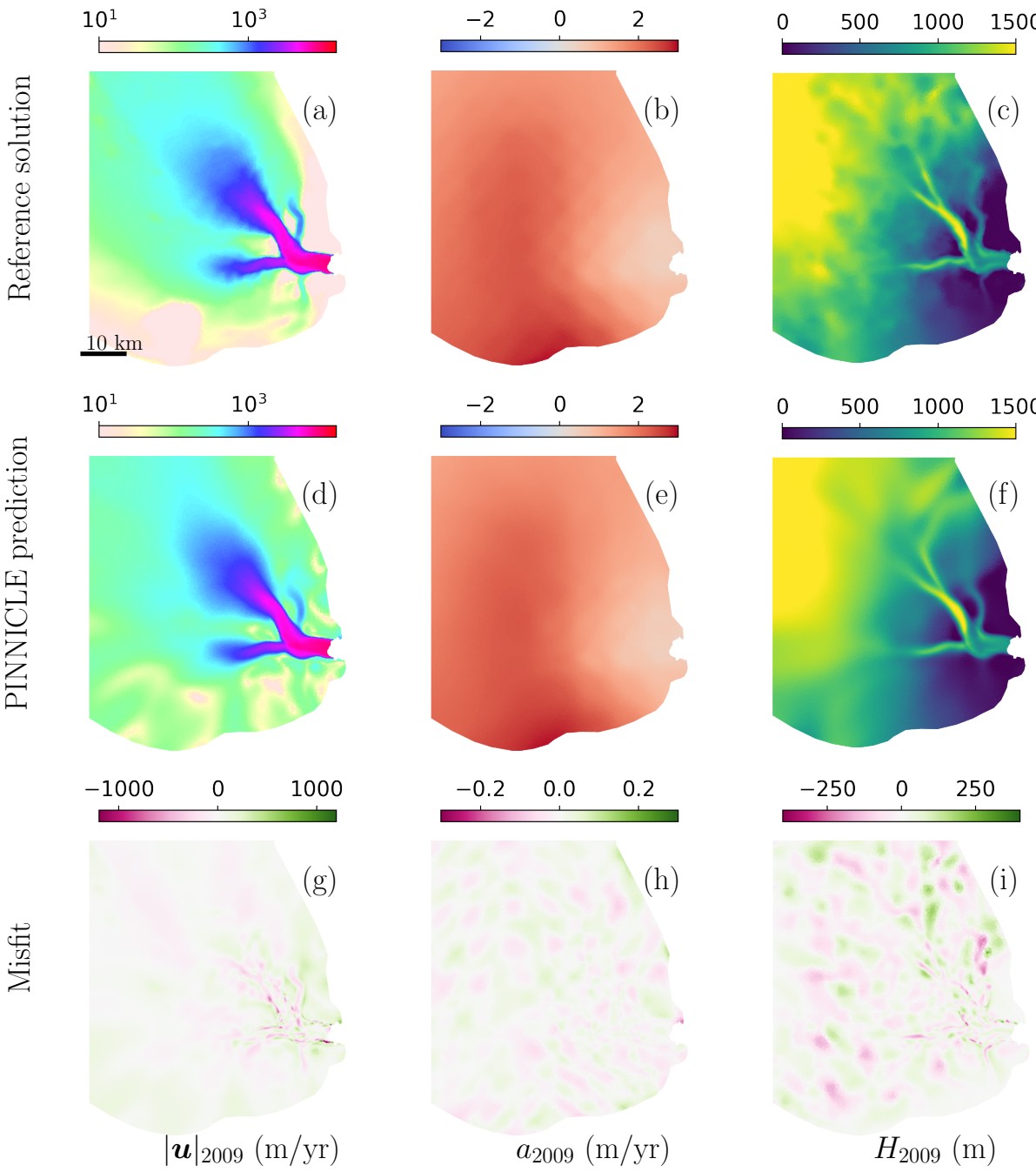

**Figure 7.** The comparison of the 'reference solution', PINNICLE predictions and misfits of the transient simulation in example 3 at the final time step $t = 2009$. (a)-(c) The 'reference solution' of surface velocity, mass balance, and ice thickness. Note that (c) is not exposed to the training. (d)-(f) The PINNICLE predictions. (g)-(i) The misfit between the 'reference solution' and the corresponding PINNICLE predictions.