# Peer review of "A Python library for solving ice sheet modeling problems using Physics Informed Neural Networks, PINNICLE v1.0"

_EGUsphere, 2025_

## Referee Comment (RC1)

**Review of the manuscript:** A Python library for solving ice sheet modeling problems using Physics Informed Neural Networks, PINNICLE v1.0

This manuscript presents the development of a new Python package that uses physics-informed neural networks (PINNs) to solve inverse problems in ice-sheet and ice-shelf dynamics. The authors provide a comprehensive description of their methods, including the network architecture, the physical constraints, the loss function, and the training hyperparameters, all of which are significant for successful training. The authors also provide concrete examples and technical details for using the package to solve various types of problems. Based on the provided example code, the package appears to be efficient and user-friendly. The manuscript is well-written and has a clear structure. I recommend this manuscript for publication, provided the authors address the following questions regarding novelty and reproducibility:

1. **Loss function weight:**
As mentioned in the manuscript, the PINN loss function comprises different terms, including data misfit and equation residuals. Each term requires a prefactor to weigh its contribution to the total loss. Table 2 provides the default values for these weights in PINNICLE, which show order-of-magnitude differences.

   - Regarding robustness, how can a user determine if these default values are suitable for their specific problem?

   - If adjustments are necessary, could the authors provide some rules-of-thumb for modifying these values, particularly for the equation residual weights?

   - Regarding implementation, could the authors include an example in one of Listings 1-3 demonstrating how to set weights different from the default values?

      .

2. **Fourier Feature:**
The authors mention the use of Fourier Features in PINNICLE, expressed as $f(x)=[\cos(Bx),\sin(Bx)]$, where $B$ is sampled from a Gaussian distribution $N(0,\sigma)$. According to Tancik et al., $\sigma$ is a hyperparameter that users must carefully determine to capture high-frequency features in the data without overfitting.

   - Therefore, does PINNICLE automatically select the optimal $\sigma$ value for generating the Fourier Feature net weights based on different training data, or should the user set this value manually?

   - If the latter, what is the default value of $\sigma$ in PINNICLE?

3. **Random sampling:**
In Line 197, the authors state that "...PINNICLE employs a random sampling strategy to automatically load data…".

- Regarding the random sampling strategy, does PINNICLE exclusively use a uniform distribution for random sampling, or are other options available for users to select?

- Does PINNICLE permit users to implement their own customized sampling strategies?

- Another question regarding random sampling is whether the data sampling points and the collocation points used for evaluating equation residuals are fixed throughout the training process or re-sampled at certain iteration intervals?

4. **Mass Balance:**
The mass balance equation (Equation 1) includes a net mass accumulation term, a.

- When incorporating this equation into the PINN training, does the user need to provide data for a across the entire domain, or can a be treated as a constant?

- If a can be treated as a constant, what is its default value in PINNICLE?

**5. Novelty:** Another Python package, DIFFICE-jax, which also uses PINNs to study ice dynamics, was recently published. Regarding novelty, could the authors elaborate on the differences between PINNICLE and DIFFICE-jax?

**Minor Point:**

- Figure 1 appears somewhat sparse. Consider shrinking the image or reorganizing its content to make it more compact.

**Suggested Additional Citations:**
The following relevant publications on PINNs in glaciology could also be cited:

Riel, Bryan, and Brent Minchew. "Variational inference of ice shelf rheology with physics-informed machine learning." *Journal of Glaciology* 69.277 (2023): 1167-1186.

Wang, Yongji, et al. "Deep learning the flow law of Antarctic ice shelves." *Science* 387.6739 (2025): 1219-1224.

Wang, Yongji, and Ching-Yao Lai. "DIFFICE-jax: Differentiable neural-network solver for data assimilation of ice shelves in JAX." *Journal of Open Source Software* 10.109 (2025): 7254.

---

## Referee Comment (RC2)

**Review of "A Python library for solving ice sheet modeling problems using Physics Informed Neural Networks, PINNICLE v1.0"**

Facundo Sapienza
May 2025

The manuscript contributes to the scientific literature by introducing the Python package PINNACLE for ice sheet forward and inverse modelling based on PINNs. The paper successfully addressed the technical aspects behind the implementation of these techniques. The manuscript is well written and represents an important contribution to the literature that fits perfectly well the scope of the journal.

I strongly recommend the publication of the manuscript in GMD after the minor revisions and comments here presented are addressed. If I decided to divide my comments between major and minor, please interpret my major comments as rather being intermediate. I think it would be important to address these points in the revised version of the manuscript, but this does not shade the overall quality of the work.

All comments, both major and minor, are aimed to improve the quality of the paper or to elucidate some of my doubts or questions.

**Major Comments**

- **Automatic Differentiation machinery.** The manuscript does not discuss much about the automatic differentiation techniques used for training the PINN. There are mentions of the use of different AD backeds in the abstract and introductions, and there is a mention of the use of "backpropagation" in line 74. This last mention made me conclude (maybe wrongly) that all the derivatives in the code are computed using reverse AD. The reason I am bringing this up is because it is well known that derivatives with respect to a few parameters are more efficiently being computed using forward AD rather than reverse AD (backpropagation). In the case of a PINN, this implies that derivatives with respect to the input layer (t, x, y) should be computed with forward AD and derivatives with respect to the parameters of the neural network should be computed with reverse AD. The combination of these two types of derivatives results in nested AD codes that may be more complicated to implement but they are definitely more both memory and time efficient.

  It will be important in the manuscript to clarify how these two types of derivatives are computed. Furthermore, this has a major impact on the performance of the code, reason why I will encourage the authors to add a mention to this in Section 9 since this directly has an impact on the performance of the method.

- **Data section.** There is no description of the data that will be used in the model (from a generic perspective). Up to this point in the manuscript, it is not clear if these data correspond to ice thickness, velocities, etc; How are they sampled? What are the temporal-spatial resolutions of these? etc. This gets more clear with the examples, but up to this point it is unclear what the data are. I will recommend starting this section with an introduction to the data products or types considered for the simulation rather than the data format of these hypothetical datasets. The

second paragraph [136-141] does not really fit in this section, and I would recommend moving it to Section 6 which addresses the topic of loss functions.

- **Confusion about the use of weights in the losses and the normalization of them.** I am a bit confused with Table 2: are all these quantities normalized in the code? I think there is some confusion in the use of weights. One one side, there are the weights of the different terms in the loss functions that need to be calibrated with some model selection technique (in this case, by running multiple simulations and selecting the best one). On the other hand, the weights mentioned in the context of Table 2 seem to refer to the weights of individual physical variables. These two are different in nature, and the use of one does not exclude the other. I find the discussion in this section to be mixing these two notions of weighthing and scaling.

**Minor Comments**

- [30] No need to capitalize "Partial Differential Equation".

- [33] The authors comment that classical methods are difficult in part due to the demand of regularization. However, this is also the case with PINNs and many other techniques. The physical term in the loss function is nothing else than a regularization term. The use of regularization is common in most inverse modelling techniques, both based in numerical solvers and in machine learning techniques. I am aware the authors are familiar with this concept, as it is clear from line 46. However, when introduced in the context of classical techniques it seems to me it is overemphasized the negative effect of having to use regularization techniques. I will suggest expanding on the difficulties of regularization in classical settings, or mention that this challenge is common to both families of methods.

- [39] Please check the consistency between capitalizing or not Physics Informed Neural Network in the manuscript (in the current version, both occur). Personally, I don't think the method should be capitalized, but it is up to the authors.

- I am not sure Figure No1 really helps much with the reading of the manuscript in its current format. The figure does not present much content, and it is very similar to the figure in Cheng (2024) and to the later figures in the manuscript, which make a better point in explaining the design of the PINN. I will suggest to abstract this figure a bit more to really represent a high-level scheme of the tool (e.g., with illustrations showing how the data and physics may look) or directly to remove the figure.

- [72] Maybe change "depending on the governing equations" -> "for time-dependent PDEs".

- [79] The manuscript states " the training procedure is to minimize the loss function, such that the output of the neural network satisfies the governing PDEs and also matches the data provided". This is not true for PINNs, at least not in principle. PINNs include the differential equation as a soft constraint in an optimization procedure, and there is no guarantee (in principle) that the residual of the physical term loss is zero (or close enough to zero). I think this is an important point that should be clear in the manuscript (which is covered in more detail in Cheng 2024).

- [98] I will suggest moving the reference to the end of the line so reference and acronym are not mixed.

- [Eq 1 & 2] Please check the consistency between the use of u, v and $\bar{u}$, $\bar{v}$ between equations. I imagine in Eq 2 the velocities are the vertical averaged ones, but this is not clear from the text.

- [102] H has been already introduced in [91].

- [110] Please notice that the friction law derived in Weertman 1957 actually corresponds to m=2, no m=3. If variations and corrections to the original Weertman theory have been made, I will try to include one of these references if m=3 is going to be the standard value (see for example Section 3.2 in Law et. al (2025) "What is glacier sliding?").

- [125] I am not sure "b" (bed) has been introduced in the text. Probably a good place to introduce it is after Equation 5, where it would be important to show that the normalized depth factor goes from zero to one between bed and surface.

- [139] There may be cases where $\epsilon$ may be selected larger than machine precision.

- Caption in Table 1 is missing a period.

- [153] Is the denormalization also done with a min-max scaling?

- [155] I am confused about the normalization here used. All physical quantities are defined in the same units systems, as they should be. On the other hand, the neural network inputs and outputs dimensionless quantities, and dimensionalization of them is done in a pre- and post-processing step. However, this does not necessarily mean that there is no further need to normalize the physical equations. For example, Equation 6 still has dimensional quantities involved (e.g, rho and g). Can the author clarify on this point, both if they were trying to imply something different or if they effectively do a dimensional analysis of the equations. This point may be related to my major concern regarding Table 2.

- [159] It is clear here that the operation cos(Bx) is applied coordinate-wise on the vector Bx. Since this is not the notation used in the rest of the paper, it may be worth clarifying.

- [163] The example of the loss function depicted in Equation 9 is very simplistic, and at this point in the manuscript still leaves uncertainty about what exactly is the optimization problem to be solved. This is probably familiar to people familiar with PINNs, but maybe not to the general audience. I think it is ok to leave Equation 9 as it is, but I will suggest adding a sentence in this section clarifying that more concrete examples of losses will be provided in the later sections including examples. I am more of a fan of the introduction of the losses as it is presented in the PINNACLE documentation (https://pinnicle.readthedocs.io/en/latest/training/lossfunctions.html), where the authors use an example to show how a general loss function would look like. This presentation also helps illustrate better the role of the weight/hyperparameter in the physical terms in the loss function, which is not clear as it is in Equation 9 right now.

- [174] Can the authors clarify how they quantify "best performance" over the different experiments? Is this based in some other metric or based on the different contributions to the loss

term to be of the same order? This last point is very important, since it is usually a challenge in any model involving multiple losses. Would the authors provide some references here that can help strengthen this point?

- [182] "to evolve alongside advances in physics-informed machine learning in the future" -> "to evolve alongside future advances in physics-informed machine learning"

- [191] I would suggest adding the references to the software packages mentioned in the manuscript (e.g., JAX). It does always help to cite open source libraries :)

- [195] If the random sampling strategy is important for the learning task, I will encourage the authors to include this mention in the "Loss function" section rather than in the "structure of the package". Even when this can be mentioned in this later section too, it is important that a random sampling strategy is used for training. This is a form of implicit regularization applied to the optimization problem.

- [201] The authors mentioned the use of Hammersley sequence sampling which produces a random set of collocation points. My understanding of this method after a quick read based on the provided reference (Wong 1997) is that Hammersley sampling generates a deterministic set of points, not random. Random versions of this method exist though (see for example Section 8.6.2 of the book "Sampling and Reconstruction"). Can the authors please clarify if the sampling method is deterministic or not, and if not provide a reference to understand the sampling procedure? Notice that the use of a deterministic or random sampling technique for the collocation method can have an impact in the learning task. (I am glad the authors mentioned this in the manuscript and they motivate me to better understand this technique, so please take this comment as a suggested improvement of someone who cares about the little details in the methods.) Furthermore, are the polygons generated with these sampled points? Do they use a different technique?

- [217] The authors clarify the point that the friction coefficient C is not used as data. However, I still find it confusing that the friction is included in the data box in Figure 3. Considering a picture is worth a thousand words, I will suggest the authors re-arrange Figure 3 to not display the friction coefficient in the data box. In this way, it won't be necessary to clarify this in the manuscript. Furthermore, it is a bit confusing that in Figure 4 the friction coefficient C is plotted alongside other observable quantities at the same level. I will suggest moving the figures with C to the right of the figure and maybe plot a line in between this and the plots for H, u, and s, to make it clear that this is not an observation.

- [223] Missing "+" symbol in Eq 10, beginning of second line.

- [231] Is it possible to run the experiment for longer? This will show reproducibility of the results compared with Cheng (2024).

- Missing period at the end of caption for Listing 1, 2, and 3.

- [243] I don't think it is necessary nor helps the reader to include the name of the data files in the manuscript (both PIG.mat and BC.mat). Unless the authors are trying to make a point by naming the files, I will suggest removing such references.

- [297] I am not sure about this last point in the performance section. What do the authors mean by "effectively resolving inversion problems with smoother, physically consistent solutions"? Solutions of the classical adjoint method will actually have strict physical consistent solutions, at differences of PINNs that provide a relaxation of the numerical solution of the underlying differential equation.

- I was taking a quick look at the Python package and this looks great, very nice work. Just to improve the presentation of the work, please let me suggest further adding a link to the docs in the code availability section and an executable example if possible (for example, a colab or Binder link where the simplest of the model can be executed). This is of course not really required, but I think it will be an interesting further contribution to the manuscript.

- Comment about the listings 1, 2, and 3. I like the initiative of the authors to show how the software can be used from the user perspective and the different parameters of the models that can be specified inside PINNACLE. However, I am not convinced that the three listings do successfully communicate this point, since the scripts shown are rather specific and communicate rather the effort of the authors in writing a very modular library. I think it is ok to keep one of the listings for illustration, but I don't really see a win in having the three of them in the main manuscrupt. Instead, I will suggest having a table with the different parameters than can be customize in the model (the ones currently in the lines of code in the listing plus some others) and also reference the reader in the manuscript to a concrete example (e.g., something contain in a Jupyter notebook or Python script). Following my previous comment, even better if this example can be directly executed using colab or Binder (even a very small toy example will do the job). This is a suggestion as it is up to the authors to decide what to do with this comment.

---

## Author Comment (AC2)

**Response to the reviewer 1**

May 31, 2025

**General comments**

This manuscript presents the development of a new Python package that uses physics-informed neural networks (PINNs) to solve inverse problems in ice-sheet and ice-shelf dynamics. The authors provide a comprehensive description of their methods, including the network architecture, the physical constraints, the loss function, and the training hyperparameters, all of which are significant for successful training. The authors also provide concrete examples and technical details for using the package to solve various types of problems. Based on the provided example code, the package appears to be efficient and user-friendly. The manuscript is well-written and has a clear structure. I recommend this manuscript for publication, provided the authors address the following questions regarding novelty and reproducibility:

**Response:** We appreciate the reviewer for their thoughtful review and positive feedback on our manuscript. The comments and suggestions have been carefully addressed below.

1. **Loss function weight**: As mentioned in the manuscript, the PINN loss function comprises different terms, including data misfit and equation residuals. Each term requires a prefactor to weigh its contribution to the total loss. Table 2 provides the default values for these weights in PINNICLE, which show order-of-magnitude differences.

   - Regarding robustness, how can a user determine if these default values are suitable for their specific problem?

     **Response:** Thank you for this important question. As discussed in the manuscript, the selection of appropriate weights in the PINN loss function remains an open research topic, with little theoretical guidance currently available in the literature. The default weights we proposed in Table 2 were determined empirically through over 15,000 numerical experiments, and have been shown to provide robust performance in a variety of test cases. However, we acknowledge that the optimal weights can vary depending on the specific characteristics of the problem. Therefore, we recommend that users assess the suitability of these default weights by monitoring the convergence

history and the individual errors of each term during training, ensuring that the contributions from each term remain balanced for their specific problem. PINNICLE facilitates this process by recording the full training history, which can be used to adjust the weights accordingly.

- If adjustments are necessary, could the authors provide some rules-of-thumb for modifying these values, particularly for the equation residual weights?
  **Response:** Indeed, as mentioned in line 174 of the manuscript, our general rule-of-thumb is to adjust the weights so that the contributions from each term are approximately of the same order of magnitude—typically on the order of 1. We recognize that this recommendation could be more explicit, and we will revise the manuscript to clearly highlight this principle for the reader.

- Regarding implementation, could the authors include an example in one of Listings 1-3 demonstrating how to set weights different from the default values?
  **Response:** Thank you for this suggestion. We have provided detailed instructions for adjusting weights in our online documentation (`https://pinnicle.readthedocs.io/en/latest/training/lossfunctions.html`). We will add this in example 1 to demonstrate how users can customize weights directly.

2. **Fourier Feature**: The authors mention the use of Fourier Features in PINNICLE, expressed as $f(x) = [\cos(Bx), \sin(Bx)]$, where B is sampled from a Gaussian distribution $N(\theta, \sigma)$. According to Tancik et al., $\sigma$ is a hyperparameter that users must carefully determine to capture high-frequency features in the data without overfitting.

   (a) Therefore, does PINNICLE automatically select the optimal $\sigma$ value for generating the Fourier Feature net weights based on different training data, or should the user set this value manually?
       **Response:** This parameter has to be set manually, as demonstrated in example 2.

   (b) If the latter, what is the default value of $\sigma$ in PINNICLE?
       **Response:** The default value in PINNICLE is a constant 10, and by default the FFT is turned off.

3. **Random Sampling**: In Line 197, the authors state that "...PINNICLE employs a random sampling strategy to automatically load data..."

   (a) Regarding the random sampling strategy, does PINNICLE exclusively use a uniform distribution for random sampling, or are other options available for users to select?
       **Response:** Currently, PINNICLE only supports uniform random sampling using *numpy.random.choice*. However, we plan to provide more options for sampling strategies in future versions.

(b) Does PINNICLE permit users to implement their own customized sampling strategies?

**Response:** Yes, since PINNICLE is open source, advanced users can implement and integrate their own customized sampling strategies as needed.

(c) Another question regarding random sampling is whether the data sampling points and the collocation points used for evaluating equation residuals are fixed throughout the training process or re-sampled at certain iteration intervals?

**Response:** By default, data sampling points and collocation points are fixed throughout the training process. However, we are planning to provided experimental resampling functions for advanced users who wish to re-sample at specified intervals.

4. **Mass Balance**: The mass balance equation (Equation 1) includes a net mass accumulation term, a.

(a) When incorporating this equation into the PINN training, does the user need to provide data for a across the entire domain, or can a be treated as a constant?

**Response:** Yes, the mass balance term $\dot{a}$ is treated as a spatially varying variable in PINNICLE, as demonstrated in Figure 7 and 8.

(b) If a can be treated as a constant, what is its default value in PINNICLE?

**Response:** Unfortunately, because $\dot{a}$ is an important variable in the mass balance equation, it is only considered as an output variable of the PINN and there is no default value provided by PINNICLE.

5. **Novelty**: Another Python package, DIFFICE-jax, which also uses PINNs to study ice dynamics, was recently published. Regarding novelty, could the authors elaborate on the differences between PINNICLE and DIFFICE-jax?

**Response:** We are indeed aware of DIFFICE-jax, and have had close collaborations with its authors. The two packages were developed almost simultaneously. In terms of differences, PINNICLE is designed to support multiple backends, including TensorFlow, PyTorch, and JAX, which offers users flexibility depending on their preferences, hardware compatibility, and experience.

**Minor point**

Figure 1 appears somewhat sparse. Consider shrinking the image or reorganizing its content to make it more compact.

**Response:** Thank you for this suggestion. We decide to remove Figure 1.

**Suggested Additional Citations**

The following relevant publications on PINNs in glaciology could also be cited:
**Response:** Change has been made.

Riel, Bryan, and Brent Minchew. "Variational inference of ice shelf rheology with physics-informed machine learning. " Journal of Glaciology 69.277 (2023): 1167-1186.
Wang, Yongji, et al. 1219-1224. "Deep learning the flow law of Antarctic ice shelves. " Science 387.6739 (2025):
Wang, Yongji, and Ching-Yao Lai. "DIFFICE-jax: Differentiable neural-network solver for data assimilation of ice shelves in JAX. " Journal of Open Source Software 10.109 (2025): 7254.

---

## Author Comment (AC3)

**Response to the reviewer 2**

May 31, 2025

**General comments**

The manuscript contributes to the scientific literature by introducing the Python package PINNICLE for ice sheet forward and inverse modelling based on PINNs. The paper successfully addressed the technical aspects behind the implementation of these techniques. The manuscript is well written and represents an important contribution to the literature that fits perfectly well the scope of the journal. I strongly recommend the publication of the manuscript in GMD after the minor revisions and comments here presented are addressed. If I decided to divide my comments between major and minor, please interpret my major comments as rather being intermediate. I think it would be important to address these points in the revised version of the manuscript, but this does not shade the overall quality of the work. All comments, both major and minor, are aimed to improve the quality of the paper or to elucidate some of my doubts or questions.

**Response:** We appreciate the reviewer's thorough and thoughtful comments. Thank you for your positive feedback and for highlighting the overall quality and relevance of our work.

**Major Comments**

- **Automatic Differentiation machinery.** The manuscript does not discuss much about the automatic differentiation techniques used for training the PINN. There are mentions of the use of different AD backeds in the abstract and introductions, and there is a mention of the use of "backpropagation" in line 74. This last mention made me conclude (maybe wrongly) that all the derivatives in the code are computed using reverse AD. The reason I am bringing this up is because it is well known that derivatives with respect to a few parameters are more efficiently being computed using forward AD rather than reverse AD (backpropagation). In the case of a PINN, this implies that derivatives with respect to the input layer (t, x, y) should be computed with forward AD and derivatives with respect to the parameters of the neural network should be computed with reverse AD. The combination of these two types of derivatives results

in nested AD codes that may be more complicated to implement but they are definitely more both memory and time efficient.

It will be important in the manuscript to clarify how these two types of derivatives are computed. Furthermore, this has a major impact on the performance of the code, reason why I will encourage the authors to add a mention to this in Section 9 since this directly has an impact on the performance of the method.

**Response:** Thank you for raising this important point. We indeed use reverse-mode AD throughout the entire package. We acknowledge that, as the reviewer points out, using a mixed approach can be more computationally efficient in some cases. However, at present, the underlying backend library (DeepXDE) does not yet support this separable PINN approach. This feature is still under development in DeepXDE (see https://github.com/lululxvi/deepxde/pull/1776), and we have decided to wait for its full support before implementing it in PINNICLE. We will clarify this in the revised manuscript as suggested.

- **Data section** Data section. There is no description of the data that will be used in the model (from a generic perspective). Up to this point in the manuscript, it is not clear if these data correspond to ice thickness, velocities, etc; How are they sampled? What are the temporal-spatial resolutions of these? etc. This gets more clear with the examples, but up to this point it is unclear what the data are. I will recommend starting this section with an introduction to the data products or types considered for the simulation rather than the data format of these hypothetical datasets. The second paragraph [136-141] does not really fit in this section, and I would recommend moving it to Section 6 which addresses the topic of loss functions.

  **Response:** Thank you for highlighting this point. We understand that the current title "Data" may be misleading. This section is intended to describe the "data module" within the PINNICLE framework (as depicted in Figure 1 and referred to as "Model Data" in Figure 2), not the actual data used in our examples. Since PINNICLE is a general purpose package, it does not come with built-in datasets. The specific datasets we used in our examples were sourced from previous research and ISSM tutorials, and we will add appropriate citations to make this clear. Regarding the second paragraph, we will keep it in the place, since the data misfit functions belong to the data module. Finally, to avoid confusion, we will consistently use the term "Data module" throughout the manuscript.

- **Confusion about the use of weights in the losses and the normalization of them** I am a bit confused with Table 2: are all these quantities normalized in the code? I think there is some confusion in the use of weights. One one side, there are the weights of the different terms in the loss functions that need to be calibrated with some model selection technique (in this case, by running multiple simulations and selecting

the best one). On the other hand, the weights mentioned in the context of Table 2 seem to refer to the weights of individual physical variables. These two are different in nature, and the use of one does not exclude the other. I find the discussion in this section to be mixing these two notions of weighting and scaling.

**Response:** Thanks for highlighting this point of confusion. In PINNI-CLE, normalization is applied only to the inputs and outputs of the neural network itself. This normalization is done automatically to map all input values to a range typically between -1 and 1, and output values from [-1, 1] to its original values, which is a common practice to improve convergence and numerical stability in neural networks. However, when evaluating the PDE residuals, which involves nonlinear terms and multiple physical variables, we compute these residuals using the output of the neural network in their original SI unit system (i.e., after denormalization). As a result, the weights listed in Table 2 serve to balance the contributions of each physical variable and residual in the total loss function. This ensures that the different physical terms have comparable magnitudes and that no single term dominates the optimization process. To avoid confusion, we will add a clear explanation in the neural network section distinguishing between input/output normalization as an architectural feature of the network and the physical variable weights in the loss function, which ensure balanced training across physical processes.

**Minor Comments**

- [30] No need to capitalize "Partial Differential Equation"
  **Response:** Change has been made.

- [33] The authors comment that classical methods are difficult in part due to the demand of regularization. However, this is also the case with PINNs and many other techniques. The physical term in the loss function is nothing else than a regularization term. The use of regularization is common in most inverse modelling techniques, both based in numerical solvers and in machine learning techniques. I am aware the authors are familiar with this concept, as it is clear from line 46. However, when introduced in the context of classical techniques it seems to me it is overemphasized the negative effect of having to use regularization techniques. I will suggest expanding on the difficulties of regularization in classical settings, or mention that this challenge is common to both families of methods.
  **Response:** Thank you for this insightful comment. We will rephrase this sentence.

- [39] Please check the consistency between capitalizing or not Physics Informed Neural Network in the manuscript (in the current version, both occur). Personally, I don't think the method should be capitalized, but it is up to the authors.

**Response:** Thanks for pointing out this. We decided to capitalize it because it is an common acronym, but this point can be revised during copy editing based on the journal's preference.

- I am not sure Figure 1 really helps much with the reading of the manuscript in its current format. The figure does not present much content, and it is very similar to the figure in Cheng (2024) and to the later figures in the manuscript, which make a better point in explaining the design of the PINN. I will suggest to abstract this figure a bit more to really represent a high-level scheme of the tool (e.g., with illustrations showing how the data and physics may look) or directly to remove the figure.
  **Response:** Thank you for this suggestion. We decide to remove Figure 1.

- [72] Maybe change "depending on the governing equations" -> "for time-dependent PDEs".
  **Response:** Thank you for this suggestion. Our intent was to emphasize that the inputs can include both spatial and temporal variables depending on the specific PDEs being solved, not only for time-dependent PDEs. To avoid confusion, we will revise the sentence to clarify this point more explicitly.

- [79] The manuscript states " the training procedure is to minimize the loss function, such that the output of the neural network satisfies the governing PDEs and also matches the data provided". This is not true for PINNs, at least not in principle. PINNs include the differential equation as a soft constraint in an optimization procedure, and there is no guarantee (in principle) that the residual of the physical term loss is zero (or close enough to zero). I think this is an important point that should be clear in the manuscript (which is covered in more detail in Cheng 2024).
  **Response:** Thanks, we will make it more clear.

- [98] I will suggest moving the reference to the end of the line so reference and acronym are not mixed
  **Response:** Change has been made.

- [Eq 1 & 2] Please check the consistency between the use of u, v and $\bar{u}, \bar{v}$ between equations. I imagine in Eq 2 the velocities are the vertical averaged ones, but this is not clear from the text.
  **Response:** Change has been made.

- [102] H has been already introduced in [91].
  **Response:** Change has been made.

- [110] Please notice that the friction law derived in Weertman 1957 actually corresponds to m=2, no m=3. If variations and corrections to the original Weertman theory have been made, I will try to include one of these references if m=3 is going to be the standard value (see for example

Section 3.2 in Law et. al (2025) "What is glacier sliding?").

**Response:** Thank you for bringing this to our attention. As noted in Cuffey and Paterson (2010) (page 236), the use of $m = 3$ reflects improvements to the original formulation proposed by Fowler (1979, 1981). We will clarify this point in the manuscript.

- [125] I am not sure "b" (bed) has been introduced in the text. Probably a good place to introduce it is after Equation 5, where it would be important to show that the normalized depth factor goes from zero to one between bed and surface.
  **Response:** Change has been made.

- [139] There may be cases where $\epsilon$ may be selected larger than machine precision.
  **Response:** Yes, but in this case, we just try to avoid $d_i = 0$.

- Caption in Table 1 is missing a period.
  **Response:** Change has been made.

- [153] Is the denormalization also done with a min-max scaling?
  **Response:** Yes, We will clarify this point in the revised manuscript.

- [155] I am confused about the normalization here used. All physical quantities are defined in the same units systems, as they should be. On the other hand, the neural network inputs and outputs dimensionless quantities, and dimensionalization of them is done in a pre- and post-processing step. However, this does not necessarily mean that there is no further need to normalize the physical equations. For example, Equation 6 still has dimensional quantities involved (e.g, rho and g). Can the author clarify on this point, both if they were trying to imply something different or if they effectively do a dimensional analysis of the equations. This point may be related to my major concern regarding Table 2.
  **Response:** We have clarified this point in our response to the major concern raised by the reviewer.

- [159] It is clear here that the operation cos(Bx) is applied coordinate-wise on the vector Bx. Since this is not the notation used in the rest of the paper, it may be worth clarifying.
  **Response:** Change has been made.

- [163] The example of the loss function depicted in Equation 9 is very simplistic, and at this point in the manuscript still leaves uncertainty about what exactly is the optimization problem to be solved. This is probably familiar to people familiar with PINNs, but maybe not to the general audience. I think it is ok to leave Equation 9 as it is, but I will suggest adding a sentence in this section clarifying that more concrete examples of losses will be provided in the later sections including examples. I am more of a fan of the introduction of the losses as it is presented in the PINNACLE documentation (`https://pinnicle.readthedocs.io/en/latest/training/`

`lossfunctions.html`), where the authors use an example to show how a general loss function would look like. This presentation also helps illustrate better the role of the weight/hyperparameter in the physical terms in the loss function, which is not clear as it is in Equation 9 right now.
**Response:** Thank you for this helpful suggestion. We will add a clarifying sentence to this section, explicitly guiding the reader to the detailed examples and loss function formulations presented in Section 9.

- [174] Can the authors clarify how they quantify "best performance" over the different experiments? Is this based in some other metric or based on the different contributions to the loss term to be of the same order? This last point is very important, since it is usually a challenge in any model involving multiple losses. Would the authors provide some references here that can help strengthen this point?
**Response:** thank you for this question. The detailed approach is presented in Cheng et al. (2024), where an L-curve analysis is conducted to identify the optimal balance. We will add a sentence to describe this analysis in the revised manuscript to strengthen this point.

- [182] "to evolve alongside advances in physics-informed machine learning in the future" evolve alongside future advances in physics-informed machine learning"
**Response:** Change has been made.

- [191] I would suggest adding the references to the software packages mentioned in the manuscript (e.g., JAX). It does always help to cite open source libraries :)
**Response:** Change has been made.

- [195] If the random sampling strategy is important for the learning task, I will encourage the authors to include this mention in the "Loss function" section rather than in the "structure of the package". Even when this can be mentioned in this later section too, it is important that a random sampling strategy is used for training. This is a form of implicit regularization applied to the optimization problem.
**Response:** We agree with this suggestion and will add a clarifying sentence.

- [201] The authors mentioned the use of Hammersley sequence sampling which produces a random set of collocation points. My understanding of this method after a quick read based on the provided reference (Wong 1997) is that Hammersley sampling generates a deterministic set of points, not random. Random versions of this method exist though (see for example Section 8.6.2 of the book "Sampling and Reconstruction"). Can the authors please clarify if the sampling method is deterministic or not, and if not provide a reference to understand the sampling procedure? Notice that the use of a deterministic or random sampling technique for the collocation method can have an impact in the learning task. (I am glad the

authors mentioned this in the manuscript and they motivate me to better understand this technique, so please take this comment as a suggested improvement of someone who cares about the little details in the methods.) Furthermore, are the polygons generated with these sampled points? Do they use a different technique?
**Response:** Thank you for pointing this out. The Hammersley sequence sampling we used is quasi-random and deterministic, not truly random. We will revise the manuscript to clarify this distinction.

- [217] The authors clarify the point that the friction coefficient C is not used as data. However, I still find it confusing that the friction is included in the data box in Figure 3. Considering a picture is worth a thousand words, I will suggest the authors re-arrange Figure 3 to not display the friction coefficient in the data box. In this way, it won't be necessary to clarify this in the manuscript. Furthermore, it is a bit confusing that in Figure 4 the friction coefficient C is plotted alongside other observable quantities at the same level. I will suggest moving the figures with C to the right of the figure and maybe plot a line in between this and the plots for H, u, and s, to make it clear that this is not an observation.
**Response:** Thank you for your suggestion. While we understand the potential confusion, we would like to keep the current presentation of Figure 3 as is. This is because, in this particular example, we do use the basal friction coefficient $C$ on the boundary of the domain, and it is therefore included in the data box to indicate its role in the setup. However, we will revise the manuscript text to make this distinction more explicit and reduce any possible confusion for the reader.

- [223] Missing "+" symbol in Eq 10, beginning of second line.
**Response:** Change has been made.

- [231] Is it possible to run the experiment for longer? This will show reproducibility of the results compared with Cheng (2024)
**Response:** Thank you for your question. We have intentionally presented a shorter version of the experiment here to provide a concise and accessible example for readers and potential users. For a longer run, the results would indeed be the same as those reported in Cheng (2024).

- Missing period at the end of caption for Listing 1, 2, and 3.
**Response:** Change has been made.

- [243] I don't think it is necessary nor helps the reader to include the name of the data files in the manuscript (both PIG.mat and BC.mat). Unless the authors are trying to make a point by naming the files, I will suggest removing such references.
**Response:** Thank you for your suggestion. While we agree that including the data file names may not be strictly necessary, we decided to keep them in the manuscript to illustrate that they represent different datasets.

- [297] I am not sure about this last point in the performance section. What do the authors mean by "effectively resolving inversion problems with smoother, physically consistent solutions"? Solutions of the classical adjoint method will actually have strict physical consistent solutions, at differences of PINNs that provide a relaxation of the numerical solution of the underlying differential equation.

  **Response:** We argue that traditional numerical methods also impose physical constraints in a weak or approximate sense, depending on the specific method used. Similarly, PINNs are designed as global approximation methods, which inherently introduce some numerical errors while approximating the solution. Nevertheless, both approaches aim to produce physically meaningful solutions within the limitations of their respective formulations.

- I was taking a quick look at the Python package and this looks great, very nice work. Just to improve the presentation of the work, please let me suggest further adding a link to the docs in the code availability section and an executable example if possible (for example, a colab or Binder link where the simplest of the model can be executed). This is of course not really required, but I think it will be an interesting further contribution to the manuscript.

  **Response:** Thanks for the kind feedback and helpful suggestions. We have recently updated our online documentation and have also provided a Docker image in the documentation to help users run the package. However, because platforms like Colab and Binder do not provide permanent, citable links, which are required by Copernicus, we have decided not to include those. Instead, we have ensured that the documentation and code are permanently archived together on Zenodo, so users have stable access.

- Comment about the listings 1, 2, and 3. I like the initiative of the authors to show how the software can be used from the user perspective and the different parameters of the models that can be specified inside PINNACLE. However, I am not convinced that the three listings do successfully communicate this point, since the scripts shown are rather specific and communicate rather the effort of the authors in writing a very modular library. I think it is ok to keep one of the listings for illustration, but I don't really see a win in having the three of them in the main manuscript. Instead, I will suggest having a table with the different parameters than can be customize in the model (the ones currently in the lines of code in the listing plus some others) and also reference the reader in the manuscript to a concrete example (e.g., something contain in a Jupyter notebook or Python script). Following my previous comment, even better if this example can be directly executed using colab or Binder (even a very small toy example will do the job). This is a suggestion as it is up to the authors to decide what to do with this comment.

  **Response:** Thank you for this suggestion, which aligns with a similar comment by the Executive Editor, Juan A. Añel. We have now uploaded

all necessary data, including the inputs for all examples, data used for training, and the neural network weights after training. These resources are available in the PINNICLE/examples/ folder of our package, archived with a permanent DOI: https://doi.org/10.5281/zenodo.15178900.

---

## Author Response (AR2)

Dear Dr. Räss,

Thank you for this excellent suggestion. Indeed, PINNICLE already supports the ability to treat the basal friction coefficient $C$ as a completely free parameter by excluding it from the observational dataset. We have now added a clarification in Example 1 describing how to activate this functionality and updated the manuscript accordingly to reflect this capability.

Thank you for your support and guidance.

Best regards,
Cheng Gong, Mansa Krishna, Mathieu Morlighem